# Exact learning dynamics of deep linear networks with prior knowledge

**Lukas Braun** [†,1]
lukas.braun@psy.ox.ac.uk

**Clémentine C. J. Dominé** [†,2]
clementine.domine.20@ucl.ac.uk

**James E. Fitzgerald** [3]
fitzgeraldj@janelia.hhmi.org

**Andrew M. Saxe** [2,4,5]
a.saxe@ucl.ac.uk

## Abstract

Learning in deep neural networks is known to depend critically on the knowledge embedded in the initial network weights. However, few theoretical results have precisely linked prior knowledge to learning dynamics. Here we derive exact solutions to the dynamics of learning with rich prior knowledge in deep linear networks by generalising Fukumizu's matrix Riccati solution [1]. We obtain explicit expressions for the evolving network function, hidden representational similarity, and neural tangent kernel over training for a broad class of initialisations and tasks. The expressions reveal a class of task-independent initialisations that radically alter learning dynamics from slow non-linear dynamics to fast exponential trajectories while converging to a global optimum with identical representational similarity, dissociating learning trajectories from the structure of initial internal representations. We characterise how network weights dynamically align with task structure, rigorously justifying why previous solutions successfully described learning from small initial weights without incorporating their fine-scale structure. Finally, we discuss the implications of these findings for continual learning, reversal learning and learning of structured knowledge. Taken together, our results provide a mathematical toolkit for understanding the impact of prior knowledge on deep learning.

## 1 Introduction

A hallmark of human learning is our exquisite sensitivity to prior knowledge: what we already know affects how we subsequently learn [2]. For instance, having learned about the attributes of nine animals, we may learn about the tenth more quickly [3, 4, 5, 6]. In machine learning, the impact of prior knowledge on learning is evident in a range of paradigms including reversal learning [7], transfer learning [8, 9, 10, 11], continual learning [12, 13, 14], curriculum learning [15], and meta learning [16]. One form of prior knowledge in deep networks is the initial network state, which is known to strongly impact learning dynamics [17, 18, 19]. Even random initial weights of different variance can yield qualitative shifts in network behaviour between the *lazy* and *rich* regimes [20], imparting distinct inductive biases on the learning process. More broadly, rich representations such as those obtained through pretraining provide empirically fertile inductive biases for subsequent

---

† First authors, random order
1. Department of Experimental Psychology, University of Oxford, Oxford, United Kingdom
2. Gatsby Computational Neuroscience Unit, University College London, London, United Kingdom
3. Howard Hughes Medical Institute, Janelia Research Campus, Ashburn, USA
4. Sainsbury Wellcome Centre, University College London, London, United Kingdom
5. CIFAR Azrieli Global Scholar, CIFAR, Toronto, Canada

36th Conference on Neural Information Processing Systems (NeurIPS 2022).

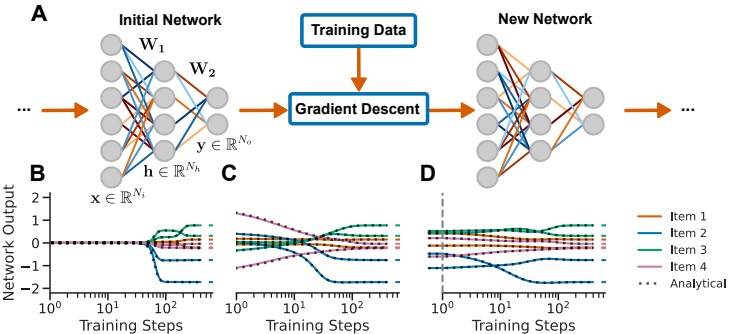

Figure 1: Learning with prior knowledge. **A** In our setting, a deep linear network with $N_i$ input, $N_h$ hidden and $N_o$ output neurons is trained from a particular initialisation using gradient descent. **B-D** Network output for an example task over training time when starting from **B** small random weights, **C** large random weights, and **D** the weights of a previously learned task. The dynamics depend in detail on the initialisation. Solid lines indicate simulations, dotted lines indicate the analytical solutions we derive in this work.

fine-tuning [21]. Yet while the importance of prior knowledge to learning is clear, our theoretical understanding remains limited, and fundamental questions remain about the implicit inductive biases of neural networks trained from structured initial weights. A better understanding of the impact of initialisation on gradient-based learning may lead to improved pretraining schemes and illuminate pathologies like catastrophic forgetting in continual learning [22].

Here, we address this gap by deriving exact solutions to the dynamics of learning in deep linear networks as a function of network initialisation, revealing an intricate and systematic dependence. We consider the setting depicted in Fig. 1A, where a network is trained with standard gradient descent from a potentially complex initialisation. When trained on the same task, different initialisations can radically change the network's learning trajectory (Fig. 1B-D). Our approach, based on a matrix Riccati formalism [1], provides explicit analytical expressions for the network output over time (Fig. 1B-D dotted). While simple, deep linear networks have a non-convex loss landscape and have been shown to recapitulate several features of nonlinear deep networks while retaining mathematical tractability.

## 1.1 Contributions

- We derive an explicit solution for the gradient flow of the network function, internal representational similarity, and finite-width neural tangent kernel of over- and under-complete two-layer deep linear networks for a rich class of initial conditions (Section 3).

- We characterise a set of random initial network states that exhibit fast, exponential learning dynamics and yet converge to *rich* neural representations. Dissociating fast and slow learning dynamics from the *rich* and *lazy* learning regimes (Section 4).

- We analyse how weights dynamically align to task-relevant structure over the course of learning, going beyond prior work that has assumed initial alignment (Section 5).

- We provide exact solutions to continual learning dynamics, reversal learning dynamics and to the dynamics of learning and revising structured representations (Section 6).

## 1.2 Related work

Our work builds on analyses of deep linear networks [23, 1, 17, 24, 10, 25, 26, 27], which have shown that this simple model nevertheless has intricate fixed point structure and nonlinear learning dynamics reminiscent of phenomena seen in nonlinear networks. A variety of works has analysed convergence [28, 29], generalisation [10, 30, 31], and the implicit bias of gradient descent [32, 33, 34, 35]. These works mostly considers the *tabula rasa* case of small initial random weights, for which exact solutions are known [17]. By contrast our formalism describes dynamics from a much larger class of initial conditions and can describe alignment dynamics that do not occur in the *tabula rasa* setting. Most directly, our results build from the matrix Riccati formulation proposed by [1]. Connecting this

formulation and matrix factorisation problems yields a better characterisation of the convergence rate [27]. We extend and refine the matrix Riccati result to obtain the dynamics of over- and under-complete networks; to obtain numerically stable forms of the matrix equations; and to more explicitly reveal the impact of initialisation.

A line of theoretical research has considered online learning dynamics in teacher-student settings [36, 37, 38], deriving ordinary differential equations for the average learning dynamics even in nonlinear networks. However, solving these equations requires numerical integration. By contrast, our approach provides explicit analytical solutions for the more restricted case of deep linear networks.

Other approaches for analysing deep network dynamics include the Neural Tangent Kernel (NTK) [39, 40, 41] and the mean field approach [42, 43, 44]. While the former can describe nonlinear networks but not the learning dynamics of hidden representations, the later yields a description of representation learning dynamics in wide networks in terms of a partial differential equation. Our work is similar in seeking a subset of more tractable models that are amenable to analysis, but we focus on the impact of initialisation on representation learning dynamics and explicit solutions.

A large body of work has investigated the effect of different random initialisations on learning in deep networks. The role of initialisation in the vanishing gradient problem and proposals for better initialisation schemes have been illuminated by several works drawing on the central limit theorem [45, 17, 46, 18, 47], reviewed in [48, 19, 49]. These approaches typically guarantee that gradients do not vanish at the start of learning, but do not analytically describe the resulting learning trajectories. Influential work has shown that network initialisation variance mediates a transition from *rich* representation learning to *lazy* NTK dynamics [20], which we analyse in our framework.

## 2 Preliminaries and setting

Consider a supervised learning task in which input vectors $\mathbf{x}_n \in \mathbb{R}^{N_i}$ from a set of $P$ training pairs $\{(\mathbf{x}_n, \mathbf{y}_n)\}_{n=1...P}$ have to be associated with their target output vectors $\mathbf{y}_n \in \mathbb{R}^{N_o}$. We learn this task with a two-layer linear network model (Fig. 1A) that produces the output prediction

$$\hat{\mathbf{y}}_n = \mathbf{W}_2 \mathbf{W}_1 \mathbf{x}_n, \tag{1}$$

with weight matrices $\mathbf{W}_1 \in \mathbb{R}^{N_h \times N_i}$ and $\mathbf{W}_2 \in \mathbb{R}^{N_o \times N_h}$, where $N_h$ is the number of hidden units. The network's weights are optimised using full batch gradient descent with learning rate $\eta$ (or respectively time constant $\tau = 1/\eta$) on the mean squared error loss

$$\mathcal{L}(\hat{\mathbf{y}}, \mathbf{y}) = \frac{1}{2} \left\langle ||\hat{\mathbf{y}} - \mathbf{y}||^2 \right\rangle, \tag{2}$$

where $\langle \cdot \rangle$ denotes the average over the dataset. The input and input-output correlation matrices of the dataset are

$$\tilde{\mathbf{\Sigma}}^{xx} = \frac{1}{P} \sum_{n=1}^{P} \mathbf{x}_n \mathbf{x}_n^T \in \mathbb{R}^{N_i \times N_i} \quad \text{and} \quad \tilde{\mathbf{\Sigma}}^{yx} = \frac{1}{P} \sum_{n=1}^{P} \mathbf{y}_n \mathbf{x}_n^T \in \mathbb{R}^{N_o \times N_i}. \tag{3}$$

Finally, the gradient optimisation starts from an initialisation $\mathbf{W}_2(0), \mathbf{W}_1(0)$. Our goal is to understand the full time trajectory of the network's output and internal representations as a function of this initialisation and the task statistics.

Our starting point is the seminal work of Fukumizu [1], which showed that the gradient flow dynamics could be written as a matrix Riccati equation with known solution. In particular, defining

$$\mathbf{Q} = \begin{bmatrix} \mathbf{W}_1^T \\ \mathbf{W}_2 \end{bmatrix} \quad \text{and} \quad \mathbf{F} = \begin{bmatrix} 0 & (\tilde{\mathbf{\Sigma}}^{yx})^T \\ \tilde{\mathbf{\Sigma}}^{yx} & 0 \end{bmatrix}, \tag{4}$$

the continuous time dynamics of the matrix $\mathbf{Q}\mathbf{Q}^T$ from initial state $\mathbf{Q}(0)$ is

$$\mathbf{Q}\mathbf{Q}^T(t) = e^{\mathbf{F}\frac{t}{\tau}} \mathbf{Q}(0) \left[ \mathbf{I} + \frac{1}{2} \mathbf{Q}(0)^T \left( e^{\mathbf{F}\frac{t}{\tau}} \mathbf{F}^{-1} e^{\mathbf{F}\frac{t}{\tau}} - \mathbf{F}^{-1} \right) \mathbf{Q}(0) \right]^{-1} \mathbf{Q}(0)^T e^{\mathbf{F}\frac{t}{\tau}}, \tag{5}$$

if the following four assumptions hold:

**Assumption 2.1** *The dimensions of the input and target vectors are identical, that is $N_i = N_o$.*

**Assumption 2.2** *The input data is whitened, that is $\tilde{\mathbf{\Sigma}}^{xx} = \mathbf{I}$.*

**Assumption 2.3** *The network's weight matrices are zero-balanced at the beginning of training, that is $\mathbf{W}_1(0)\mathbf{W}_1(0)^T = \mathbf{W}_2(0)^T\mathbf{W}_2(0)$. If this condition holds at initialisation, it will persist throughout training [17, 24].*

**Assumption 2.4** *The input-output correlation of the task and the initial state of the network function have full rank, that is $\mathrm{rank}(\tilde{\mathbf{\Sigma}}^{xy}) = \mathrm{rank}(\mathbf{W}_2(0)\mathbf{W}_1(0)) = N_i = N_o$. This implies that the network is not bottlenecked, i.e. $N_h \geq \min(N_i, N_o)$.*

For completeness, we include a derivation of this solution in Appendix A.

Rather than tracking the weights' dynamics directly, this approach tracks several key statistics collected in the matrix

$$\mathbf{Q}\mathbf{Q}^T = \begin{bmatrix} \mathbf{W}_1^T\mathbf{W}_1(t) & \mathbf{W}_1^T\mathbf{W}_2^T(t) \\ \mathbf{W}_2\mathbf{W}_1(t) & \mathbf{W}_2\mathbf{W}_2^T(t) \end{bmatrix}, \tag{6}$$

which can be separated into four quadrants with intuitive meaning: the off-diagonal blocks contain the network function

$$\hat{\mathbf{Y}}(t) = \mathbf{W}_2\mathbf{W}_1(t)\mathbf{X}, \tag{7}$$

while the on-diagonal blocks contain the correlation structure of the weight matrices. These permit calculation of the temporal evolution of the network's internal representations including the task-relevant representational similarity matrices (RSM) [50], i.e. the kernel matrix $\phi(x)^T\phi(x')$, of the neural representations in the hidden layer

$$\mathrm{RSM}_I = \mathbf{X}^T\mathbf{W}_1^T\mathbf{W}_1(t)\mathbf{X}, \quad \mathrm{RSM}_O = \mathbf{Y}^T(\mathbf{W}_2\mathbf{W}_2^T(t))^+\mathbf{Y}, \tag{8}$$

where $+$ denotes the pseudoinverse; and the network's finite-width neural tangent kernel [39, 40, 41]

$$\mathrm{NTK} = \mathbf{I}_{N_o} \otimes \mathbf{X}^T\mathbf{W}_1^T\mathbf{W}_1(t)\mathbf{X} + \mathbf{W}_2\mathbf{W}_2^T(t) \otimes \mathbf{X}^T\mathbf{X}, \tag{9}$$

where $\mathbf{I}$ is the identity matrix and $\otimes$ is the Kronecker product. For a derivation of these quantities see Appendix B. Hence, the solution in Equation 5 describes important aspects of network behaviour.

However, in this form, the solution has several limitations. First, it relies on general matrix exponentials and inverses, which are a barrier to explicit understanding. Second, when evaluated numerically, it is often unstable. And third, the equation is only valid for equal input and output dimensions. In the following section we address these limitations.

**Implementation and simulation** Simulation details are in Appendix H. Code to replicate all simulations and plots are available online[1] under a *GPLv3* license and requires <6 hours to execute on a single AMD Ryzen 5950x.

## 3 Exact learning dynamics with prior knowledge

In this section we derive an exact and numerically stable solution for $\mathbf{Q}\mathbf{Q}^T$ that better reveals the learning dynamics, convergence behaviour and generalisation properties of two-layer linear networks with prior knowledge. Further, we alter the equations to be applicable to equal and unequal input and output dimensions, overcoming Assumption 2.1).

To place the solution in a more explicit form, we make use of the compact singular value decomposition. Let the compact singular value decomposition of the initial network function and the input-output correlation of the task be

$$\mathrm{SVD}(\mathbf{W}_2(0)\mathbf{W}_1(0)) = \mathbf{U}\mathbf{S}\mathbf{V}^T \quad \text{and} \quad \mathrm{SVD}(\tilde{\mathbf{\Sigma}}^{yx}) = \tilde{\mathbf{U}}\tilde{\mathbf{S}}\tilde{\mathbf{V}}^T. \tag{10}$$

Here, $\mathbf{U}$ and $\tilde{\mathbf{U}} \in \mathbb{R}^{N_o \times N_m}$ denote the left singular vectors, $\mathbf{S}$ and $\tilde{\mathbf{S}} \in \mathbb{R}^{N_m \times N_m}$ the square matrix with ordered, non-zero eigenvalues on its diagonal and $\mathbf{V}$ and $\tilde{\mathbf{V}} \in \mathbb{R}^{N_i \times N_m}$ the corresponding right singular vectors. For unequal input-output dimensions ($N_i \neq N_o$) the right and left singular vectors are therefore not generally square and orthonormal. Accordingly, for the case $N_i > N_o$, we define $\tilde{\mathbf{U}}_\perp \in \mathbb{R}^{N_o \times (N_o - N_i)}$ as a matrix containing orthogonal column vectors that complete the basis, i.e., make $\begin{bmatrix} \tilde{\mathbf{U}} & \tilde{\mathbf{U}}_\perp \end{bmatrix}$ orthonormal. Conversely, we define $\tilde{\mathbf{V}}_\perp \in \mathbb{R}^{N_i \times (N_i - N_o)}$ for the case of $N_i > N_o$.

---

[1]https://github.com/saxelab/deep-linear-networks-with-prior-knowledge

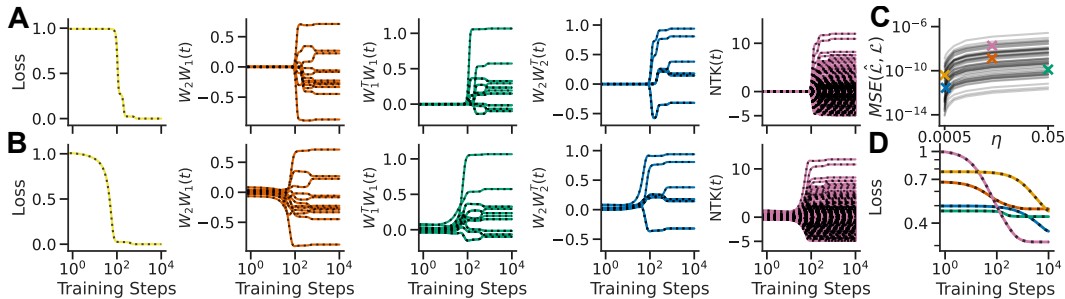

Figure 2: Exact learning dynamics **A** The temporal dynamics of the numerical simulation (coloured lines) of the loss, network function, correlation of input and output weights and the NTK (columns 1-5 respectively) are exactly matched by the analytical solution (black dotted lines) for small initial weight values and **B** large initial weight values. **C** Each line shows the deviation of the analytical loss $\hat{\mathcal{L}}$ from the numerical loss $\mathcal{L}$ for one of $n = 50$ networks with random architecture and training data (details in Appendix H) across a range of learning rates $\eta \in [0.05, 0.0005]$. The deviation mutually decreases with the learning rate. **D** Numerical and analytical learning curves for five randomly sampled example networks (coloured x in C).

**Assumption 3.1** *Define* $\mathbf{B} = \mathbf{U}^T\tilde{\mathbf{U}} + \mathbf{V}^T\tilde{\mathbf{V}}$ *and* $\mathbf{C} = \mathbf{U}^T\tilde{\mathbf{U}} - \mathbf{V}^T\tilde{\mathbf{V}}$. $\mathbf{B}$ *is non-singular.*

**Theorem 3.1** *Under the assumptions of whitened inputs, 2.2, zero-balanced weights 2.3, full rank 2.4, and* $\mathbf{B}$ *non-singular 3.1, the temporal dynamics of* $\mathbf{QQ}^T$ *are*

$$
\begin{aligned}
\mathbf{QQ}^T(t) = \mathbf{Z}\Big[ & 4e^{-\tilde{\mathbf{S}}\frac{t}{\tau}}\mathbf{B}^{-1}\mathbf{S}^{-1}(\mathbf{B}^T)^{-1}e^{-\tilde{\mathbf{S}}\frac{t}{\tau}} + \left(\mathbf{I} - e^{-2\tilde{\mathbf{S}}\frac{t}{\tau}}\right)\tilde{\mathbf{S}}^{-1} \\
& - e^{-\tilde{\mathbf{S}}\frac{t}{\tau}}\mathbf{B}^{-1}\mathbf{C}\left(e^{-2\tilde{\mathbf{S}}\frac{t}{\tau}} - \mathbf{I}\right)\tilde{\mathbf{S}}^{-1}\mathbf{C}^T(\mathbf{B}^T)^{-1}e^{-\tilde{\mathbf{S}}\frac{t}{\tau}} \\
& + 4\frac{t}{\tau}e^{-\tilde{\mathbf{S}}\frac{t}{\tau}}\mathbf{B}^{-1}(\mathbf{V}^T\tilde{\mathbf{V}}_\perp\tilde{\mathbf{V}}_\perp^T\mathbf{V} + \mathbf{U}^T\tilde{\mathbf{U}}_\perp\tilde{\mathbf{U}}_\perp^T\mathbf{U})(\mathbf{B}^T)^{-1}e^{-\tilde{\mathbf{S}}\frac{t}{\tau}}\Big]^{-1}\mathbf{Z}^\mathbf{T}
\end{aligned}
\tag{11}
$$

*with*

$$
\mathbf{Z} = \begin{bmatrix}
\tilde{\mathbf{V}}\left(\mathbf{I} - e^{-\tilde{\mathbf{S}}\frac{t}{\tau}}\mathbf{C}^T(\mathbf{B}^T)^{-1}e^{-\tilde{\mathbf{S}}\frac{t}{\tau}}\right) + 2\tilde{\mathbf{V}}_\perp\tilde{\mathbf{V}}_\perp^T\mathbf{V}(\mathbf{B}^T)^{-1}e^{-\tilde{\mathbf{S}}\frac{t}{\tau}} \\
\tilde{\mathbf{U}}\left(\mathbf{I} + e^{-\tilde{\mathbf{S}}\frac{t}{\tau}}\mathbf{C}^T(\mathbf{B}^T)^{-1}e^{-\tilde{\mathbf{S}}\frac{t}{\tau}}\right) + 2\tilde{\mathbf{U}}_\perp\tilde{\mathbf{U}}_\perp^T\mathbf{U}(\mathbf{B}^T)^{-1}e^{-\tilde{\mathbf{S}}\frac{t}{\tau}}
\end{bmatrix}.
\tag{12}
$$

For a proof of Theorem 3.1 please refer to Appendix C.

With this solution we can calculate the exact temporal dynamics of the loss, network function, RSMs and NTK (Fig. 2A, B). As the solution contains only negative exponentials, it is numerically stable and provides high precision across a wide range of learning rates and network architectures (Fig. 2C, D).

We note that a solution for the weights $\mathbf{W}_1(t)$ and $\mathbf{W}_2(t)$, i.e., $\mathbf{Q}(t)$, can be derived up to a time varying orthogonal transformation as demonstrated in Appendix C. Further, as time-dependent variables only occur in matrix exponentials of diagonal matrices of negative sign, the network approaches a steady state solution.

**Theorem 3.2** *Under the assumptions of Theorem 3.1, the network function converges to the global minimum* $\tilde{\mathbf{U}}\tilde{\mathbf{S}}\tilde{\mathbf{V}}^T$ *and acquires a rich task-specific internal representation, that is* $\mathbf{W}_1^T\mathbf{W}_1 = \hat{\mathbf{V}}\tilde{\mathbf{S}}\hat{\mathbf{V}}^T$ *and* $\mathbf{W}_2\mathbf{W}_2^T = \tilde{\mathbf{U}}\tilde{\mathbf{S}}\tilde{\mathbf{U}}^T$.

The proof of Theorem 3.2 is in Appendix C. We now turn to several implications of these results.

## 4  Rich and lazy learning regimes and generalisation

Recent results have shown that large deep networks can operate in qualitatively distinct regimes that depend on their weight initialisations [20, 51], the so called *rich* and *lazy* regimes. In the *rich* regime, learning dynamics can be highly nonlinear and lead to task-specific solutions thought

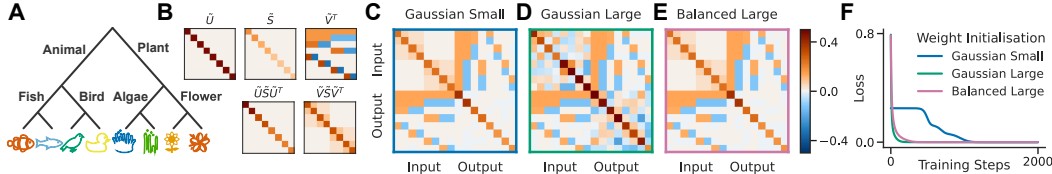

Figure 3: Rich and lazy learning. **A** Semantic learning task, **B** SVD of the input-output correlation of the task (top) and the respective RSMs (bottom). Rows and columns in the SVD and RSMs are identically ordered as the order of items in the hierarchical tree. **C** Final $\mathbf{QQ}^T$ matrices after training converged when initialised from random small weights, **D** random large weights (note how the upper left and lower right quadrant differ from the task's RSMs) and **E** large zero-balanced weights. **F** Learning curves for the three different initialisations as in C (green), D (pink) and E (blue). While both large weight initialisations lead to fast exponential learning curves, the small weight initialisation leads to a slow step-like decay of the loss.

to lead to favourable generalisation properties [20, 25, 51]. By contrast, the *lazy* regime exhibits simple exponential learning dynamics and exploits high-dimensional nonlinear projections of the data produced by the initial random weights, leading to task-agnostic representations that attain zero training error but possibly lower generalisation performance [39, 40, 41]. Traditionally, the *rich* and *lazy* learning regimes have been respectively linked to low and high variance initial weights (relative to the network layer size).

To illustrate these phenomena, we consider a semantic learning task in which a set of living things have to be linked to their position in a hierarchical structure (Fig. 3A) [17]. The representational similarity of the input of the task ($\tilde{\mathbf{V}}\tilde{\mathbf{S}}\tilde{\mathbf{V}}^T$) reveals its inherent structure (Fig. 3B). For example, the representations of the two fishes are most similar to each other, less similar to birds and least similar to plants. Likewise, the representational similarity of the task's target values ($\tilde{\mathbf{U}}\tilde{\mathbf{S}}\tilde{\mathbf{U}}^T$) reveals the primary groups among which items are organised. As a consequence, one can for example predict from an object being a fish that it is an animal and from an object being a plant that it is not a bird. Reflecting these structural relationships in internal representations can allow the *rich* regime to generalise in ways the *lazy* regime cannot. Crucially, $\mathbf{QQ}^T(t)$ contains the temporal dynamics of the weights' representational similarity and therefore can be used to study if a network finds a *rich* or *lazy* solution.

When training a two layer network from random small initial weights, the weights' input and output RSM (Fig. 3C, upper left and lower right quadrant) are identical to the task's structure at convergence. However, when training from large initial weights, the RSM reveals that the network has converged to a *lazy* solution (Fig. 3D). We emphasise that the network function in both cases is identical (Fig. 3C, D, lower left quadrant). And while their final loss is identical too, their learning dynamics evolve slow and step-wise in the case of small initial weights and fast and exponentially in the case of large initial weights (Fig. 3F), as predicted by previous work [20].

However, from Theorem 3.2 it directly follows that our setup is guaranteed to find a *rich* solution in which the weights' RSM is identical to the task's RSM, i.e., $\mathbf{W}_1^T\mathbf{W}_1 = \tilde{\mathbf{V}}\tilde{\mathbf{S}}\tilde{\mathbf{V}}^T$ and $\mathbf{W}_2\mathbf{W}_2^T = \tilde{\mathbf{U}}\tilde{\mathbf{S}}\tilde{\mathbf{U}}^T$. Therefore, as zero-balanced weights may be large, there exist initial states that converge to *rich* solutions while evolving as rapid exponential learning curves (Fig. 3E, F). Crucially, these initialisations are task-agnostic, in the sense that they are independent of the task structure (cf. [52]). This finding applies to any learning task with well defined input-output correlation. For additional simulations see Appendix D. Hence our equation can describe the change in dynamics from step-like to exponential with increasing weight scale, and separate this dynamical phenomenon from the structure of internal representations.

## 5 Decoupling dynamics

The learning dynamics of deep linear networks depend on the exact initial values of the synaptic weights. Previous solutions studied learning dynamics under the assumption that initial network weights are "decoupled", such that the initial state of the network and the task share the same singular vectors, i.e. that $\mathbf{U} = \tilde{\mathbf{U}}$ and $\mathbf{V} = \tilde{\mathbf{V}}$ [17]. Intuitively, this assumption means that there is no cross-coupling between different singular modes, such that each evolves independently. However, this

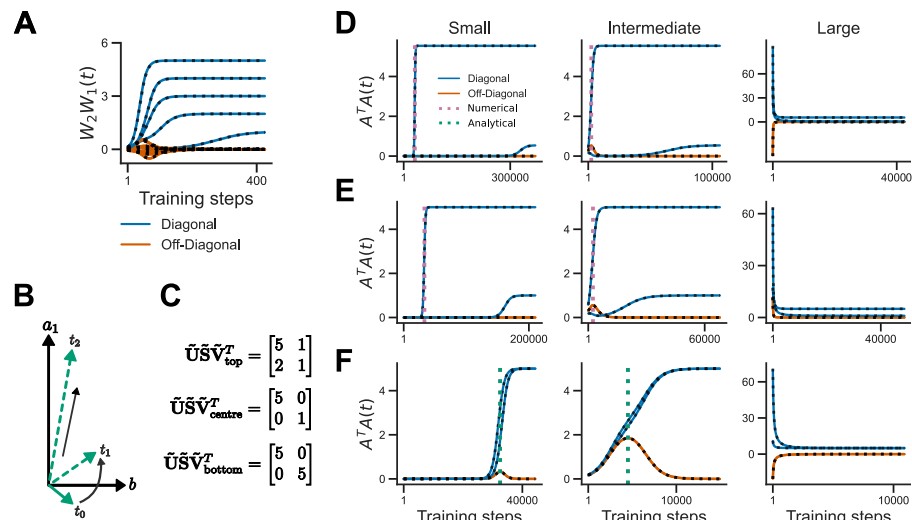

Figure 4: Decoupling dynamics. **A** Analytical (black dotted lines) and numerical (solid lines) of the temporal dynamics of the on- and off-diagonal elements of $\mathbf{A}^T\mathbf{A}$ in blue and red, respectively. **B** Schematic representation of the decoupling process. **C** Three target matrices with dense, unequal diagonal, and equal diagonal structure. **D-F** Decoupling dynamics for the top (D), middle (E), and bottom (F) tasks depicted in panel C. Row F contains analytical predictions for the time of the peak of the off-diagonal (dashed green). The network is initialised as defined in E with small, intermediate and large variance.

assumption is violated in most real-world scenarios. As a consequence, most prior work has relied on the empirical observation that learning from *tabula rasa* small initial weights occurs in two phases: First, the network's input-output map rapidly decouples; then subsequently, independent singular modes are learned in this decoupled regime. Because this decoupling process is fast when training begins from small initial weights, the learning dynamics are still approximately described by the temporal learning dynamics of the singular values assuming decoupling from the start. This dynamic has been called a *silent alignment* process [26]. Here we leverage our matrix Riccati approach to analytically study the dynamics of this decoupling process. We begin by deriving an alternate form of the exact solution that eases the analysis.

**Theorem 5.1** *Let the weight matrices of a two layer linear network be initialised by* $\mathbf{W}_1 = \mathbf{A}(0)\tilde{\mathbf{V}}^T$ *and* $\mathbf{W}_2 = \tilde{\mathbf{U}}\mathbf{A}(0)^T$, *where* $\mathbf{A}(0) \in \mathbb{R}^{N_h \times N_i}$ *is an arbitrary, invertible matrix. Then, under the assumptions of equal input-output dimensions 2.1, whitened inputs 2.2, zero-balanced weights 2.3 and full rank 2.4, the temporal dynamics of* $\mathbf{Q}\mathbf{Q}^T$ *are fully determined by*

$$\mathbf{A}^T\mathbf{A}(t) = \left[ e^{-\tilde{\mathbf{S}}\frac{t}{\tau}} \left( \mathbf{A}(0)^T\mathbf{A}(0) \right)^{-1} e^{-\tilde{\mathbf{S}}\frac{t}{\tau}} + (\mathbf{I} - e^{-2\tilde{\mathbf{S}}\frac{t}{\tau}})\tilde{\mathbf{S}}^{-1} \right]^{-1}. \tag{13}$$

For a proof of Theorem 5.1, please refer to Appendix E. We remark that this form is less general than that in Theorem 3.1, and in particular implies $\mathbf{U}\mathbf{V} = \tilde{\mathbf{U}}\tilde{\mathbf{V}}$. Here the matrix $\mathbf{A}^T\mathbf{A}$ represents the dynamics directly in the SVD basis of the task. Off-diagonal elements represent counterproductive coupling between different singular modes (for instance, $[\mathbf{A}^T\mathbf{A}]_{21}$ is the strength of connection from input singular vector 1 to output singular vector 2, which must approach zero to perform the task perfectly), while on-diagonal elements represent the coupling within the same mode (for instance, $[\mathbf{A}^T\mathbf{A}]_{11}$ is the strength of connection from input singular vector 1 to output singular vector 1, which must approach the associated task singular value to perform the task perfectly). Hence the decoupling process can be studied by examining the dynamics by which $\mathbf{A}^T\mathbf{A}$ becomes approximately diagonal.

The outer inverse in Equation 13 renders it difficult to study high dimensional networks analytically. Therefore, we focus on small networks with input and output dimension $N_i = 2$ and $N_o = 2$, for which a lengthy but explicit analytical solution is given in Appendix E. In this setting, the structure of the weight initialisation and task are encoded in the matrices

$$\mathbf{A}(0)^T\mathbf{A}(0) = \begin{bmatrix} a_1(0) & b(0) \\ b(0) & a_2(0) \end{bmatrix} \quad \text{and} \quad \tilde{\mathbf{S}} = \begin{bmatrix} s_1 & 0 \\ 0 & s_2 \end{bmatrix}, \tag{14}$$

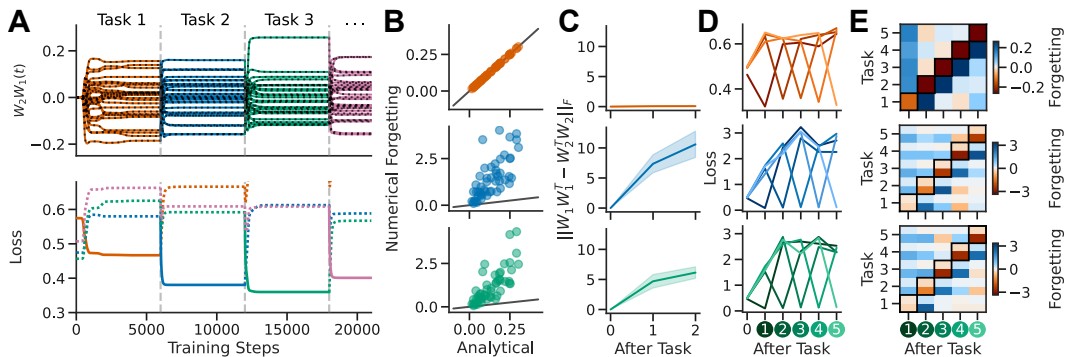

Figure 5: Continual learning. **A** Top: Network training from small zero-balanced weights on a sequence of tasks (coloured lines show simulation and black dotted lines analytical results). Bottom: Evaluation loss for tasks of the sequence (dotted) while training on the current task (solid). As the network function is optimised on the current task, the loss of other tasks increases. **B** Comparison of the numerical and analytical amount of catastrophic forgetting on a first task after training on a second task for $n = 50$ linear (red), $\tanh$ (blue) and ReLU (green) networks. **C** Weight alignment before and after training on a sequence of two tasks for $n = 50$ networks in linear (red), $\tanh$ (blue) and ReLU (green) networks. Shaded area shows $\pm$ std. **D** Evaluation loss for each of $5$ tasks during training a linear (red), $\tanh$ (blue) and ReLU (green) network. **E** Same data es in D but evaluated as relative change (i.e. amount of catastrophic forgetting). The top half of each square shows the pre-computed analytical amount of forgetting and the bottom half the numerical value.

where the parameters $a_1(0)$ and $a_2(0)$ represent the component of the initialisation that is aligned with the task, and $b(0)$ represents cross-coupling, such that taking $b(0) = 0$ recovers previously known and more restricted solutions for the decoupled case [17]. We use this setting to demonstrate two features of the learning dynamics.

**Decoupling dynamics.** First, we track decoupling by considering the dynamics of the off-diagonal element $b(t)$ (Fig. 4D-F red lines). At convergence, the off-diagonal element shrinks to zero as shown in Appendix E. However, strikingly, $b(t)$ can exhibit non-monotonic trajectories with transient peaks or valleys partway through the learning process. In particular, in Appendix E we derive the time of the peak magnitude as $t_{peak} = \frac{\tau}{4s} \ln \frac{s(s-a_1-a_2)}{a_1 a_2 - b(0)^2}$ (Fig. 4F green dotted line), which coincides approximately with the time at which the on-diagonal element is half learned. If initialised from small random weights, the off-diagonal remains near-zero throughout learning, reminiscent of the silent alignment effect [26]. For large initialisations, no peak is observed and the dynamics are exponential. At intermediate initialisations, the maximum of the off-diagonal is reached before the singular mode is fully learned (Appendix E). Intuitively, a particular input singular vector can initially project appreciably onto the wrong output singular vector, corresponding to initial misalignment. This is only revealed when this link is amplified, at which point corrective dynamics remove the counterproductive coupling, as schematised in Fig. 4B. We report further measurements of decoupling in Appendix E.

**Effect of initialisation variance.** Next, we revisit the impact of initialisation scale for the on-diagonal dynamics. As shown in Fig. 4D-F, as the initialisation variance grows the learning dynamics change from sigmoidal to exponential, possibly displaying more complex behavior at intermediate variance (Appendix E). In this simple setting we can analyse this transition in detail. Taking $s_1 = s_2 = s$ as in Fig. 4F and $|a_1(0)|, |a_2(0)|, |b(0)| \ll 1$, we recover a sigmoidal trajectory,

$$a_1(t) = \frac{sa_1(0)}{e^{\frac{-2st}{\tau}} \left[s - a_1(0) - a_2(0)\right] + a_1(0) + a_2(0)}, \tag{15}$$

while for $|a_1(0)|, |a_2(0)|, |b(0)| \gg 0$ the dynamics of the on-diagonal element $a_1$ is close to exponential (Fig. 4D-F left and right columns). We examine larger networks in Appendix E.

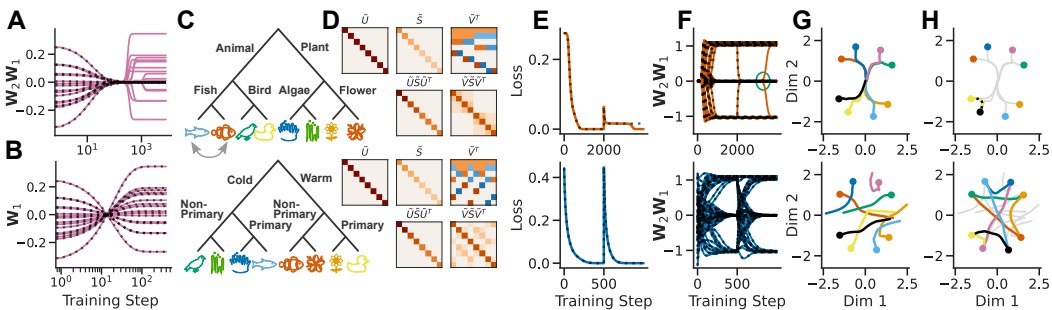

Figure 6: Reversal learning and revising structured knowledge. Scale of x-axis varies in top and bottom rows. **A** Analytical (black dotted) and numerical (solid) learning dynamics of a reversal learning task. The analytical solution gets stuck on a saddle point, whereas the numerical simulation escapes the saddle point and converges to the target. **B** In a shallow network, training on the same task as in A converges analytically (black dotted) and numerically (solid). **C** Semantic learning tasks. Revised living kingdom (top) and colour hierarchy (bottom). **D** SVD of the input-output coreelation of the tasks and respective RSMs. **E** Analytical (black dotted) and simulation (solid) loss and **F** learning dynamics of first training on the living kingdom (Fig. 3A) and subsequently on the respective task in C. The analytical solution fails for the revised animal kingdom as it gets stuck in a saddle point, while the simulation escapes the saddle (top, green circle). Initial training on the living kingdom task from large initial weights and subsequent training on the colour hierarchy have similar convergence times (bottom) **G** Multidimensional scaling (MDS) of the network function for initial training on the living kingdom task from small (top) and large initial weights (bottom). Note how despite the seemingly chaotic learning dynamics when starting form large initial weights, both simulations learn the same representation. **H** MDS of subsequent training on the respective task in C.

## 6   Applications

The solutions derived in Sections 3 and 5 provide tools to examine the impact of prior knowledge on dynamics in deep linear networks. So far we have traced general features of the behaviour of these solutions. In this section, we use this toolkit to develop accounts of several specific phenomena.

**Continual Learning**    Continual learning (see [12] for a review) and the pathology of catastrophic forgetting have long been a challenge for neural network models [22, 53, 54]. A variety of theoretical work has investigated aspects of continual learning [55, 56, 57, 58, 59]. In this setting, starting from an initial set of weights, a network is trained on a sequence of tasks with respective input-output correlations $\mathcal{T}_1 = \tilde{\Sigma}_1^{yx}, \mathcal{T}_2 = \tilde{\Sigma}_2^{yx}, \mathcal{T}_3 = \tilde{\Sigma}_3^{yx}, ....$ As shown in Fig. 5A, our dynamics immediately enable exact solutions for the full continual learning process, whereby the final state after training on one task becomes the initial network state for the next task. These solutions thus reveal the exact time course of forgetting for arbitrary sequences of tasks.

Training on later tasks can overwrite previously learned knowledge, a phenomenon known as catastrophic forgetting [22, 53, 54]. From Theorem 3.2 it follows that from any arbitrary zero-balanced initialisation 2.3, the network converges to the global optimum such that the initialisation is completely overwritten and forgetting is truly catastrophic. In particular, the loss of any other task $\mathcal{T}_i$ after training to convergence on task $\mathcal{T}_j$ is $\mathcal{L}_i(\mathcal{T}_j) = 1/2||\tilde{\Sigma}_j^{yx} - \tilde{\Sigma}_i^{yx}||_F^2 + c$, where $c$ is a constant that only depends on training data of task $\mathcal{T}_i$ (Appendix F). As a consequence, the amount of forgetting, i.e. the relative change of loss, is fully determined by the similarity structure of the tasks and thus can be fully determined for a sequence of tasks before the onset of training (Fig. 5B,E, Appendix F). For example, the amount of catastrophic forgetting in task $\mathcal{T}_a$, when training on task $\mathcal{T}_c$ after having trained the network on task $\mathcal{T}_b$ is $\mathcal{L}_a(\mathcal{T}_c) - \mathcal{L}_a(\mathcal{T}_b)$. As expected, our results depend on our linear setting and $\tanh$ or $\mathrm{ReLU}$ nonlinearities can show different behaviour, typically increasing the amount of forgetting (Fig. 5B,D,E). Further, in nonlinear networks, weights become rapidly unbalanced and forgetting values that are calculated before the onset of training do not predict the actual outcome (Fig. 5B-E). In summary, our results link exact learning dynamics with catastrophic forgetting and thus provide an analytical tool to study the mechanisms and potential counter measures underlying catastrophic forgetting.

**Reversal learning**    During reversal learning, pre-existing knowledge has to be relearned, overcoming a previously learned relationship between inputs and outputs. For example, reversal learning occurs when items of a class are mislabeled and later corrected. We show analytically, that reversal learning in fact does not succeed in deep linear networks (Appendix G). The pre-existing knowledge lies exactly on the separatrix of a saddle point causing the learning dynamics to converge to zero (Fig. 6A). In contrast, the learning still succeeds numerically, as any noise will perturb the dynamics off the saddle point, allowing learning to proceed (Fig. 6A). However, the dynamics still slow in the vicinity of the saddle point, providing a theoretical explanation for catastrophic slowing in deep linear networks [60]. We note that the analytical solution requires an adaptation of Theorem 3.1, as $B$ is generally not invertible in the case of reversal learning (Appendix G). Further, as is revealed by the exact learning dynamics (Appendix G), shallow networks do succeed without exhibiting catastrophic slowing during reversal learning (Fig. 6B).

**Revising structured knowledge**    Knowledge is often organised within an underlying, shared structure, of which many can be learned and represented in deep linear networks [25]. For example, spatial locations can be related to each other using the same cardinal directions, or varying semantic knowledge can be organised using the same hierarchical tree. Here, we investigate if deep linear networks benefit from shared underlying structure. To this end, a network is first trained on the three-level hierarchical tree of Section 4 (eight items of the living kingdom, each with a set of eight associated features), and subsequently trained on a revised version of the hierarchy. The revised task varies the relation of inputs and outputs while keeping the same underlying tree structure. If the revision involves swapping two neighbouring nodes on any level of the hierarchy, e.g. the identity of the two fish on the lowest level of the hierarchy (Fig. 6C, top), the task is identical to reversal learning, leading to catastrophically slowed dynamics (Fig. 6E-F, top). When training the network on a new hierarchical tree with identical items but a new set of features, like a colour hierarchy (Fig. 6C, bottom), there is no speed advantage in comparison to a random initialisation with similar initial variance (Fig. 6E-F, bottom). Importantly, from Theorem 3.2 it follows, that the learning process can be sped up significantly by initialising from large zero-balanced weights, while converging to a global minimum with identical generalisation properties as when training from small weights (Fig. 6G-H). In summary, having incorporated structured knowledge before revision does not speed up or even slows down learning in comparison to learning from random zero-balanced weights. Notably, that is despite the tasks' structure being almost identical (Fig. 3B and Fig. 6D).

## 7   Discussion

We derive exact solutions to the dynamics of learning with rich prior knowledge in a tractable model class: deep linear networks. While our results broaden the class of two-layer linear network problems that can be described analytically, they remain limited and rely on a set of assumptions (2.1-2.4). In particular, weakening the requirement that the input covariance be white and the weights be zero-balanced would enable analysis of the impact of initialisation on internal representations. Nevertheless, these solutions reveal several insights into network behaviour. We show that there exists a large set of initial values, namely zero-balanced weights 2.3, which lead to task-specific representations; and that large initialisations lead to exponential rather than sigmoidal learning curves. We hope our results provide a mathematical toolkit that illuminates the complex impact of prior knowledge on deep learning dynamics.

### Acknowledgments and Disclosure of Funding

L.B. was supported by the Woodward Scholarship awarded by Wadham College, Oxford and the Medical Research Council [MR/N013468/1]. C.D. and A.S. were supported by the Gatsby Charitable Foundation (GAT3755). Further, A.S. was supported by a Sir Henry Dale Fellowship from the Wellcome Trust and Royal Society (216386/Z/19/Z) and the Sainsbury Wellcome Centre Core Grant (219627/Z/19/Z). A.S. is a CIFAR Azrieli Global Scholar in the Learning in Machines & Brains program. J.F. was supported by the Howard Hughes Medical Institute.

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
