# OpenReview forum: "Exact learning dynamics of deep linear networks with prior knowledge"
_NeurIPS.cc/2022/Conference — NeurIPS 2022 Accept_

### Official Review · Reviewer_vKXk · 2022-07-01

**Rating:** 7
**Confidence:** 4
**Soundness:** 4 excellent
**Presentation:** 4 excellent
**Contribution:** 3 good

**Summary:**

This paper, entitled "Exact Learning Dynamics of Deep Linear Networks With Prior Knowledge", "derives exact solutions to the dynamics of learning" for deep linear networks. The authors leverage these solutions to understand how different regimes

**Questions:**

I have no additional questions, beyond those asked in the Strengths and Weaknesses section above, for the authors.

**Limitations:**

The authors do a great job making clear what the assumptions they rely on are and how they are limited.  The featuring of nonlinear networks (Fig. 6) was helpful to this regard and makes this point clear.

**Strengths And Weaknesses:**

STRENGTHS:
1. The paper is very clearly written and motivated. This is true both of the conceptual and technical discussion. The authors should be commended on their use of clear figures and their commentary on how their expressions should be interpreted.

2. The goal of the paper - to provide theoretical understanding and tools for studying how the specific initialization of networks affects their learning dynamics - is a worthy one. The authors make a substantiative step in this direction by overcoming a difficulty that had previously limited the development of theory.

3. The framework the authors develop provides novel insight. For instance, it was from the theory that weight matrices initialized with large values, that are in the rich regime and have exponential loss, was discovered. Similarly, the connection between input-output correlation and forgetting in deep linear networks was made. These illustrate the potential of the framework.

WEAKNESSES:
1. It is not clear to me why the task shown in Fig. 4A has the input and output correlation that is shown. Looking at the hierarchy, the fish are equally similar to each other as the flowers, but looking at the input correlation matrix, the fish are more similar to each other than the flowers. Similarly, I did not understand why the flowers have lower output correlation than the fish. This made understanding the results a little difficult.

2. That the alignment dynamics are different for different sized weights is very interesting (Fig. 5). But there was not explanation as to why some tasks should see transient increases in the alignment part way through learning. Is there some intuition/explanation that can be gleaned from the framework?

3. The forgetting metric used in Fig. 6 should be defined. What does negative forgetting correspond to? Additionally, in Fig. 6B, tanh networks look like they have considerable difference in the analytical and numerical forgetting, but in Fig. 6C, this differences look quite small. Indeed, it seems like ReLU networks have larger discrepancy when looking at Fig. 6C, but that they have smaller discrepancy when looking at Fig. 6B. Is this a matter of the color scheme? Or are the two subpanels computing different things?

4. I found it a little hard to keep track of all the nice findings the authors had (possibly because there framework "bounced around" as applied to multiple different ideas). The impact of the paper would be increased if a table (or clear discussion in Sec. 7) was included that really spelled out exactly the theoretical contributions that were made.

MINOR POINTS:
1. The referencing of figures, equations, and assumptions varies throughout the paper (eq., Eq., equation / Fig 5D, Fig. 5D, figure 5D, 5D / assumption 2.3, 2.3). These should be made consistent

2. There is a typo in Eq. 8. It should be $S^{-1}$ and not $A^{-1}$ (this is correct in the appendix just not in the main text).

3. The curves in Fig. 3C should be color coded/marked by their learning rate. As is, it is difficult to determine whether there is any pattern in the error of the approximation of the loss.

4. Is the convention $-T$ for the transpose of the inverse a common one? To me it looks like the quantity is being raised to the negative $T$ power.

5. It should be made clear in the main text that Fig. 4B-D shows the RSA computed once the network has converged.

6. Is there a typo in Fig. 5C? Is $S_{top}$ supposed to be the same as $S_{centre}$?

7. The example of learning a different hierarchy in the subsection of Sec. 6 "Revising structured knowledge" did not seem to fit in with the theme of the subsection (although I did think the result was interesting). I would suggest either dropping it or making it its own subsection.

---

> ### Author Response · Authors · 2022-08-02
> **Answer to Reviewer vKXk**
>
> We thank the reviewer for the insightful comments and are pleased the paper was clear in its writing, figures, and motivation and makes a substantive step. We have corrected all minor comments in the newly submitted version of the draft.
>
> > It is not clear to me why the task shown in Fig. 4A has the input and output correlation that is shown. Looking at the hierarchy, the fish are equally similar to each other as the flowers, but looking at the input correlation matrix, the fish are more similar to each other than the flowers. Similarly, I did not understand why the flowers have lower output correlation than the fish. This made understanding the results a little difficult.
>
> It is correct, that the sketch of the hierarchy in panel A slightly deviates from the hierarchy used to compute the results in panels B C, D and E. In the initial submission, we used an asymmetrical hierarchy with weighted labels (+/-4, +/-3 ....) as described in Appendix F. The asymmetric hierarchy has the advantage of having unique eigenvalues - which makes the visual assessment of the learning dynamics in the reversal learning case more accessible, i.e. there is no overlapping curves. However, we have now used a symmetric hierarchy (using +/-1 labels only) for Figure 4 (Figure 3 in the revised manuscript) and the asymmetric hierarchy for Figure 7 (Figure 6 in the revision) to avoid this issue. While we have updated the figure, some according updates in the appendix (e.g. simulation details) are pending. They will be included in the camera ready version.
>
> > That the alignment dynamics are different for different sized weights is very interesting (Fig. 5). But there was not explanation as to why some tasks should see transient increases in the alignment part way through learning. Is there some intuition/explanation that can be gleaned from the framework?
>
> Some intuition can be obtained from the two-by-two solution in the Appendix D. When the initialization variance is small we recover a sigmoidal trajectory for the on diagonal elements. Intuitively, in the more general case, transient increases in alignment might occur when learning a new on-diagonal element reveals misalignment, which is subsequently corrected.
>
> > The forgetting metric used in Fig. 6 should be defined. What does negative forgetting correspond to?
> Additionally, in Fig. 6B, tanh networks look like they have considerable difference in the analytical and numerical forgetting, but in Fig. 6C, this differences look quite small. Indeed, it seems like ReLU networks have larger discrepancy when looking at Fig. 6C, but that they have smaller discrepancy when looking at Fig. 6B. Is this a matter of the color scheme? Or are the two subpanels computing different things?
>
> Panel B was created by first training a network on task $i$ and then on a second task $j$. We then compared $\mathcal{L}(\mathcal{T}_j) = 1/2||{\bf \Sigma}^{yx}_j-{\bf \Sigma}^{yx}_i||_F^2$ with the empirical value $\mathcal{L}(\mathcal{T}_j) =1/2||W_2W_1X_j-{\bf \Sigma}^{yx}_i||_F^2$. Here, all values are positive. Panel C computes the `balance' of the weights ($||W_1^T W_1-W_2W_2^T||_F$). Our analysis relies on the assumption of balanced weights; therefore, a deviation from this condition indicates that the analytical solution is no longer valid. Could we kindly ask you to elaborate your question on the relation of Figure 6B and 6C?
>
> > I found it a little hard to keep track of all the nice findings the authors had (possibly because there framework "bounced around" as applied to multiple different ideas). The impact of the paper would be increased if a table (or clear discussion in Sec. 7) was included that really spelled out exactly the theoretical contributions that were made.
>
> We have added a contributions list to the appendix (Appendix H), which we will incorporate into the main text of the camera ready version with the additional space (i.e. 10th content page).

---

> > ### Comment · Reviewer_vKXk · 2022-08-05
> > **Reply to authors**
> >
> > Thank you for your response. It indeed helped clarify some of my questions.
> >
> > Thank you for correcting Fig. 4A. I now agree that the hierarchy matches the $\tilde{U}\tilde{S}\tilde{U}^T$ plot.
> >
> > Thank you for pointing out the 2$\times$2 solution in the Appendix (I believe it is in Appendix E.3 now, not Appendix D) and providing some intuition. I now feel like I have a better grasp of the behavior of the off and on diagonal elements. I did notice that there appear to be some small typos on line 638 of the Appendix.
> >
> > I apologize, I believe I mislabeled which subpanels I was confused about. What I meant to say, was that in what is now Fig. 5D, the values forgetting take ranges from -0.1 to 0.1. What does negative forgetting mean? Additionally, in Fig. 5B,the tanh network has a large spread of blue dots, suggesting that there is some difference in the analytical and numerical values. This is, of course, fine if that's how the results are. However, when I then look at Fig. 5D, it is very hard to see that the tanh network has bigger discrepancy between the analytical and numerical values, than say the ReLU network. Indeed, if anything, the ReLU network (bottom of Fig. 5D) appears to have the biggest discrepancy. Is there any chance that the ReLU and Tanh plots were swapped in either Fig. 5B or Fig. 5D? If my question still doesn't make sense, please don't hesitate to ask.

---

> > > ### Author Response · Authors · 2022-08-09
> > > **Reply to Reviewer vKXk**
> > >
> > > We would like to thank the reviewer for their clarification and further comments.
> > >
> > > > I did notice that there appear to be some small typos on line 638 of the Appendix.
> > >
> > > The typos in line 638 will be corrected for the camera-ready version.
> > >
> > > > What does negative forgetting mean?
> > >
> > > Imagine a situation in which a network is trained on a sequence of tasks $A$, $B$ and $C$ with corresponding correlation functions $\Sigma_A^{xy} = \frac{1}{P}Y_CX_C^T$, $\Sigma_B^{xy}$, $\Sigma_C^{xy}$. Let $F_A$ denote the network function after training on task $A$, i.e., $W_2W_1 = \Sigma_A^{xy}$. Let $\mathcal{F}_A$ denote the amount of forgetting by the network on task $A$, i.e., the relative change of the loss for a given task before and after training on one of the tasks. For example, the amount of forgetting on task $A$ when training on task $C$ is
> > > $$\mathcal{F}_A = \frac{1}{2P}||F_CX_A - Y_A||_F^2 - \frac{1}{2P}||F_BX_A - Y_A||_F^2,$$
> > > which is negative if the loss of task $A$ after training on task $C$ is smaller than the loss of task $A$ after training on task $B$. Please note that the forgetting for task $A$ when training on task $B$ after training on task $A$ reduced to
> > > \begin{align}
> > > \mathcal{F}_A &= \frac{1}{2P}||F_BX_A - Y_A||_F^2 - \frac{1}{2P}||F_AX_A - Y_A||_F^2\\
> > > &= \frac{1}{2}||\Sigma_B^{xy} - \Sigma_A^{xy}||,
> > > \end{align}
> > > which is guaranteed to be $\geq 0$. This is the formula, which we reported in the manuscript. As a consequence, for panel Fig. 5B, we expect the forgetting to be always positive, i.e., the loss of task $A$ always increases or remains at the same level when training on a subsequent task $B$. However, in panel Fig. 5D the loss for task $A$ may decreases when for example training on task $E$ after training on task $D$, and therefore the amount of forgetting may be negative. Based on the reviewers feedback, we have now clarified in the manuscript how forgetting is calculated in both cases and added the exact derivation of the quantity to the appendix.
> > >
> > > > However, when I then look at Fig. 5D, it is very hard to see that the tanh network has bigger discrepancy between the analytical and numerical values, than say the ReLU network.
> > >
> > > We would like to thank the reviewer for their rigorous review. Indeed, we had a typo in the code for Fig. 5D, which led to the erroneous plot. Fig. 5D has now discrepancies for both ReLU and tanh networks as one would expect from Fig. 5B.

---

### Official Review · Reviewer_rRZa · 2022-07-09

**Rating:** 6
**Confidence:** 4
**Soundness:** 4 excellent
**Presentation:** 3 good
**Contribution:** 3 good

**Summary:**

This contribution introduces a reformulation of an existing matrix Riccati formulation of learning dynamics in linear networks.
This reformulation is numerically more stable and can be used to explain various known observations.
The analysis of the dynamics is used to enable rich learning with fast alignment using high variance weights.
Finally, the insights provided by the reformulated dynamics are used to predict the learning dynamics and amount of forgetting in the context of continual learning.

**Questions:**

- Would it be possible to generalise this analysis to deeper networks?
- Can you provide sufficient conditions for the invertibility of $\mathbf{B}$?
- Could you verify that equations 7 and 8 could be simplified by using $$\mathbf{Q}\mathbf{Q}^T(t) = \mathbf{Z} \bigg[\mathbf{S}^{-1} + \mathbf{B}^T \Big(e^{\tilde{\mathbf{S}}} - \mathbf{I}\Big)\tilde{\mathbf{S}}^{-1} \mathbf{B} + \mathbf{C}^T \Big(e^{\tilde{\mathbf{S}}} - \mathbf{I}\Big)\tilde{\mathbf{S}}^{-1} \mathbf{C}\bigg]^{-1} \mathbf{Z}^T,$$ where $\mathbf{Z} = \begin{bmatrix}\tilde{\mathbf{V}} \Big(e^{\frac{1}{2} \tilde{\mathbf{S}}} \mathbf{B} + e^{-\frac{1}{2} \tilde{\mathbf{S}}} \mathbf{C} \Big) \\\\ \tilde{\mathbf{U}} \Big(e^{\frac{1}{2} \tilde{\mathbf{S}}} \mathbf{B} - e^{-\frac{1}{2} \tilde{\mathbf{S}}} \mathbf{C} \Big) \end{bmatrix},$ $\mathbf{B} = \tilde{\mathbf{V}}^T\mathbf{V} + \tilde{\mathbf{U}}^T\mathbf{U}$ and $\mathbf{C} = \tilde{\mathbf{V}}^T\mathbf{V} - \tilde{\mathbf{U}}^T\mathbf{U}$?

**Limitations:**

I think it is great that the authors added a section to the appendix to warn that they discovered a mistake in their proof for the invertibility of $\mathbf{B}$.
Apart from that, I do not think that this paper would have a direct impact on society

**Strengths And Weaknesses:**

### originality

This paper introduces a novel point of view on known dynamics.
Also, most contributions are framed in the context of relevant/related work.
However, I noticed a few sections where references could be improved:

  - The experiment with hierarchical data looks very similar to the one used in [17],
    which was introduced in (Saxe et al., 2013).
  - I believe that theorem 5.1 is more similar to the setting in [17] than the paper seems to acknowledge.
    After all, in [17,&nbsp;section 1.3] it is clearly stated that $\mathbf{W}_1 = \mathbf{R} \operatorname{diag}(\mathbf{b}) \mathbf{V}^T$ and $\mathbf{W}_2 = \mathbf{U} \operatorname{diag}(\mathbf{a}) \mathbf{R}^T$ for some orthogonal matrix $\mathbf{R}$.
    In this sense, theorem 5.1 has very similar assumptions as [17],
    in contrast to what is claimed in section 5 of the submission.

Saxe, A., McClelland, J., & Ganguli, S. Learning hierarchical category structure in deep neural networks. In Proceedings of the 35th Annual Conference of the Cognitive Science Society, 2013.

### quality

Although I like the idea, the mathematical aspects of this paper appear to be sloppy.
There is also one critical flaw: $\mathbf{B}$ might not be invertible.
Assuming that reasonable, sufficient conditions for the invertibility of $\mathbf{B}$ can be formulated, I believe most results would be correct.
However, often the proofs are not well formulated:

  - The proof for theorem 3.1 starts from equation 5.
    Therefore, theorem 3.1 would still require assumption 2.1.
  - The proof for theorem 3.2 has an unexplained gap in it.
    After all, there are no explicit expressions for $\mathbf{W}_1(t)$ and $\mathbf{W}_2$.
  - Both theorem 3.1 and a proper proof for theorem 3.2 depend on the invertibility of $\mathbf{B}$,
    which is only guaranteed for $N_i = N_o$ according to the appendix.
  - Is Assumption 3.1 an actual assumption? To me, this implies that the results only hold if $N_i \neq N_o$, which should not be necessary if I understand it correctly.
    I think it would be better to simply state that assumption 2.1 is not needed anymore.
  - I would argue that assumption 3.3 is not necessary, because it is implied by assumption 3.2.
    After all, if $N_h < \min(N_i, N_o)$, then both $\operatorname{rank}(\mathbf{W}_1(0)) \leq N_h$ and $\operatorname{rank}(\mathbf{W}_2(0)) \leq N_h$ and therefore, $\operatorname{rank}(\mathbf{W}_2(0) \mathbf{W}_1(0)) \leq \min(\operatorname{rank}(\mathbf{W}_2(0)), \operatorname{rank}(\mathbf{W}_1(0))) \leq N_h$, which leads to a contradiction, given assumption 3.2.
  - Also, _sufficient rank_ is a term I am not familiar with.
    I believe rectangular matrices are still said to be of _full rank_ if their rank is the minimum of its dimensions.
    In this sense, Assumptions 2.4 and 3.2 are equivalent.
  - The proof of theorem 3.1 introduces a limit, that (I believe) is unnecessary.
    This leads to additional terms that should always be zero, making the entire formulation harder to understand.


### clarity

Overall, the paper is a nice read and figures are clearly annotated.
However, I think there are a few concepts that would benefit from further explanation:

  - I do not understand what the RSA is.
    The reference to [49] mentions multiple steps involved in the analysis.
    However, in the text it is used as something that should be straightforward to compute.
    Figure 4A seems to indicate it should be related to the covariance,
    but I do not know if this intuition would be correct.
  - I fail to understand what it means for a solution to be _rich_ or _lazy_.
    After all, _rich_ and _lazy_ are used to describe dynamics.
  - Figure 5C introduces target matrices, which are not explained in the text.
    I would have assumed that these are the targets for $\mathbf{A}^T \mathbf{A}(t)$, but then the off-diagonal elements should converge to non-zero values.

### significance

The results of this paper could be used to establish possibly interesting connections between different types of analyses that already exist.
Moreover, the expression for the dynamics might make it possible to enable rich learning when using high variance for the initial weights.
Although it can be argued that the proposed contribution is incremental, I believe that it could be valuable for the community.
On the other hand, this approach seems to be limited to networks with a single hidden layer.

---

### Minor comments
 - on line 11: the slow dynamics are described to be _step-like_, in contrast to _exponential_.
   However, the dynamics in figure 4E are still exponential (albeit after a plateau) and I would describe the dynamics in figure 5A as _sigmoidal_.
 - on line 49-50: I tend to read _This work_ as a reference to the manuscript,
   rather than to the (variety of) work listed in the previous sentence.
 - line 63: the _but_ seems to be out of place.
 - possible typo on line 77: I would have expected $\mathbf{W}_2 \in \mathbb{R}^{N_o \times N_h}$.
 - on line 120, it might be useful to explicitly mention that this is actually the _reduced_ SVD.
 - In equation 7, I believe $\mathbf{A}^{-1}$ should be $\mathbf{S}^{-1}$.
 - on line 224, the term _troughs_ is used, but I would argue that it is more common to talk about _valleys_.
 - on lines 35 and 247, the term _catastrophic interference_ is used.
   To the best of my knowledge, this is actually called _catastrophic forgetting_, which can be considered a form of _destructive interference_.
 - on line 238, there is a reference to D, which might not belong there.
 - it would be nice to cite the conference papers rather than their pre-prints
   (e.g. [17] and [26] were presented at ICLR 2014 and 2022, respectively).

---

**update**:
Considering the revision, the answers to my questions and further discussion with the authors significantly, my original evaluation is no longer accurate.
Therefore, I decided to raise my score from 3 to 6 (as well as the soundness score from 1 to 4).

My main motivation for increasing the score:
 - A proper proof for the main theorem has been provided in the revision
 - My misunderstandings concerning the generalisation of the Fukuzima results were resolved in the discussion.
 - The main presentation issues were tackled in the revision or were promised in the discussion.

My main motivation for not increasing the score even further:
 - The analysis is inherently limited to (linear) networks with a single hidden layer.

---

> ### Author Response · Authors · 2022-08-02
> **Answer 1 to Reviewer rRZa**
>
> We thank the reviewer for the useful comments that will improve the paper. We also hope our results will help draw connections between different extant analyses. We have corrected all minor comments in the newly submitted version of the draft.
>
> > This paper introduces a novel point of view on known dynamics. Also, most contributions are framed in the context of relevant/related work. However, I noticed a few sections where references could be improved: The experiment with hierarchical data looks very similar to the one used in [17], which was introduced in (Saxe et al., 2013).
>
> Indeed these works were our direct inspiration for this dataset, we now cite Saxe et al. 2013 and further highlight this link.
>
> > I believe that theorem 5.1 is more similar to the setting in [17] than the paper seems to acknowledge. After all, in [17, section 1.3] it is clearly stated that ${\bf W_1}={\bf R}\text{diag}(b){\bf V^T}$ and ${ \bf W_2}={ \bf U} \text{diag}(a){\bf R}$t for some orthogonal matrix ${\bf R}$. In this sense, theorem 5.1 has very similar assumptions as [17], in contrast to what is claimed in section 5 of the submission.
>
> We were highly inspired by the analysis of [17], but respectfully disagree that the assumptions in Sec 5 are very similar. Ref [17], as noted by the reviewer, addresses only initialisations that are diagonal in the SVD basis (i.e. already aligned), whereas our solutions describe off-diagonal dynamics. In particular, none of the off-diagonal curves in Figure 5 (red lines, Figure 4 in the revised manuscript) can be obtained under the assumptions in [17]. These are the key results we highlight in Section 5. In more detail, the case presented in [17] is a subcase of the one we present in theorem 5.1. Namely, we initialise with  ${\bf W_1}={\bf A(0)}{\bf V^T} $ and $ { \bf W_2}={ \bf U} {\bf A(0)^T}$ giving ${\bf W_2} {\bf W_1}= { \bf U} {\bf A(0)^T}{\bf A(0)}{\bf V^T}$ where  ${ \bf U}$ and ${ \bf V} $ are the singular vectors of the target input-output correlation. In this setting ${\bf A(0)}^T{\bf A(0)}$ can have off-diagonal elements. While in  [17] initialising with  ${\bf W_1}={\bf R}\text{diag}(b){\bf V^T} $and $ { \bf W_2}={ \bf U} \text{diag}(a){\bf R^T}$  yields  $ { \bf W_2W_1}={ \bf U} \text{diag}(a)\text{diag}(b){\bf V^T}$  where $\text{diag}(a)\text{diag}(b)$ is constrained to a diagonal. In our framework, we can access the off-diagonal dynamics. This has been further highlighted in the new version.
>
> > The proof for theorem 3.1 starts from equation 5. Therefore, theorem 3.1 would still require assumption 2.1.
>
> Thank you for the close reading, one of the contributions of our work is the relaxation of assumption 2.1 (equal input-output dimensions) for theorem 3.1. We now provide a rigorous proof of theorem 3.1 in the newly submitted version (Appendix C.1), directly showing that Equations 8/9 (11/12 in the revised manuscript) satisfies the differential equation (that is, this proof does not rely on or use Eq. 5 which assumes 2.1). Because this direct proof does not provide intuition into how we arrived at this form for the solution, we leave for the reader the non-rigorous derivation that we have extensively revised based on the reviewers comments (Appendix C.2), but we will remove this if reviewers think it does not add to the paper.
>
> > The proof for theorem 3.2 has an unexplained gap in it. After all, there are no explicit expressions for $W_1$ and $W_2$.
>
> We provide an explicit proof of the limiting behaviour of the four quadrants of  ${\bf Q\bf Q}^T (t)$ in Appendix B (Appendix C.3 in the revised manuscript). Therefore, we  provide the limiting behaviour of the variables ${\bf W_2W_1(t)}$,  ${\bf W_1^T\bf W_1 } (t)$ and ${\bf W_2\bf W_2^T } (t)$. Note that we do not give a solution for the limiting behaviour of ${\bf W_1}(t)$ or ${\bf W_2}(t)$ independently, and Theorem 3.2 makes no claims regarding these.
>
> > Both theorem 3.1 and a proper proof for theorem 3.2 depend on the invertibility of ${\bf B}$, which is only guaranteed for $N_i=N_o$ according to the Appendix.
>
> In the revision of the paper (and the original submission) we state theorem 3.1 with the assumption that ${\bf B}$ is non-singular. We find that assuming the invertibility of ${\bf B}$ is minimally restrictive — all of the numerous results, insights, and observations presented in this paper hold under this assumption. Moreover, we rely on the empirical observation that ${\bf B}$ is, in all cases we have tried, invertible for un-equal input output dimensions, apart from the reversal learning case which we discuss in great detail in Appendix E (Appendix F in the revised manuscript). That is, while it would be nice to have interpretable conditions guaranteeing the invertibility of $B$, in practice this is a technical assumption that does not limit the applicability of the work to many interesting settings.
>
> (Answer continued in comment below ...)

---

> > ### Comment · Reviewer_rRZa · 2022-08-03
> > **further questions/comments (part 1)**
> >
> > Thank you for the thorough reply. I still have a few questions/comments on some of the statements and the revision.
> >
> > 1. > one of the contributions of our work is the relaxation of assumption 2.1
> >
> >    Could it be that assumption 2.1 is actually not necessary for equation 5 to hold? I scanned the Fukuzima paper and could not find where this assumption is actually used.
> >    I also do not spot where this assumption is needed in the proof included in the revision (appendix A).
> >    Finally, and maybe most importantly, the proof also works for equation 5, s.t. $\mathbf{L} = \mathbf{R}^T = e^{\mathbf{F} \frac{t}{\tau}} \mathbf{Q}(0)$ and $\mathbf{C} = \mathbf{I} + \frac{1}{2} \mathbf{Q}(0)^T \big(\ldots\big) \mathbf{Q}(0)$.
> >
> > 2.  > We now provide a rigorous proof of theorem 3.1in the newly submitted version
> >
> >    The proof shows that it is a solution, but it does not prove that the solution is unique.
> >    However, the solution from Fukuzima is provably unique.
> >    Could it be that by relaxing assumption 2.1, you lose uniqueness?
> >
> > 3. > our solutions describe off-diagonal dynamics
> >
> >    I agree that this solution would be more insightful in this regard (and an important part of this contribution in my eyes).
> >    After reading the revised manuscript, I think my confusion originated from the first few sentences in section 5:
> >
> >    > Previous solutions [17] for the exact temporal learning dynamics relied on full alignment between initial state and the target task, i.e. that $\mathbf{U} = \tilde{\mathbf{U}}$ and $\mathbf{V} = \tilde{\mathbf{V}}$.
> >    > In other words, the initial state of the network was assumed to contain part of the task’s structure before the onset of training.
> >
> >    I believe that these statements are misleading, because in theorem 5, it is still the case that $\mathbf{U} = \tilde{\mathbf{U}}$ and $\mathbf{V} = \tilde{\mathbf{V}}$.
> >    In other words, these *limitations* of [17] still apply to this work.
> >
> > 4. > $\mathbf{A}(0)^T \mathbf{A}(0)$ can have off-diagonal elements
> >
> >    This is more of a loud thought than a point of critique, but in the context of this statement, I would argue that the off-diagonal elements going to zero have more to do with orthogonalisation than with alignment to task structures. Does that make sense?
> >
> > (Answer continued in reply to comment above... )

---

> > > ### Author Response · Authors · 2022-08-07
> > > **Answer to further questions / comments (part 1)**
> > >
> > > > Could it be that assumption 2.1 is actually not necessary for equation 5 to hold? I scanned the Fukuzima paper and could not find where this assumption is actually used. I also do not spot where this assumption is needed in the proof included in the revision (appendix A). Finally, and maybe most importantly, the proof also works for equation 5 ...
> > >
> > > We note that Eqn 5 is not a solution for the unequal input/output case, as it is not defined if $F$ is singular. As we show in Eq 113-115, $F$ is necessarily singular when $N_i \neq N_o$. It has $|N_o-N_i|$ eigenvalues that are exactly zero. In generalising this solution to the unequal case, our goal is similar in spirit to the pseudoinverse--we need to understand how to handle zero eigenvalues appropriately. In this case, they contribute a time-dependent term. Dropping these new terms does not yield a solution.
> > >
> > > > The proof shows that it is a solution, but it does not prove that the solution is unique. However, the solution from Fukuzima is provably unique. Could it be that by relaxing assumption 2.1, you lose uniqueness?
> > >
> > > We conjecture that the solution remains unique, and are aiming to derive a proof of this through the full SVD. We are not in a position to answer yet, but we believe that relaxing assumption 2.1 should not affect the uniqueness of the solution.
> > >
> > > > I agree that this solution would be more insightful in this regard (and an important part of this contribution in my eyes). After reading the revised manuscript, I think my confusion originated from the first few sentences in section 5 ...
> > >
> > > Thank you, we see the issue now and will revise the first few sentences. In particular, we note that if the starting weights $W_1=AV^T$ for non-diagonal $A$, then the SVD of $W_1$ will potentially not have $V$ as the right singular vectors. That is the sense in which we meant that the task and the initialization are not aligned. We will revise as follows: "Previous solutions [17] for the exact temporal learning dynamics relied on the initial state of the weights and the target task sharing certain singular vectors (i.e. that U =  ̃U and V =  ̃V), and differing only in singular values. In other words, the initial state of the network was assumed to contain part of the task’s structure before the onset of training. Here we consider dynamics when initial weights are not diagonalized by the task SVD."
> > >
> > > We also share some dissatisfaction with the term `alignment.' We used it to highlight the relationship to prior literature on the silent alignment effect. However we also note that sending off diagonal elements to zero does not imply that $A^TA$ becomes an orthogonal matrix, another possible source of confusion. We propose using the term 'decoupling' to indicate the fact that, when all off diagonal terms are zero, there is no crosstalk between different singular modes. We will also clarify in the text that this implies that the colums of A are orthogonal (i.e. their dot product is zero but they are potentially not unit norm).

---

> > > > ### Comment · Reviewer_rRZa · 2022-08-08
> > > > **Thank you and sorry for the long discussion**
> > > >
> > > > I will try to address any further comments I have in this reply to reduce the amount of emails
> > > >
> > > > 1. > We note that Eqn 5 is not a solution for the unequal input/output case, as it is not defined if is singular.
> > > >
> > > >    This exactly what I was missing to understand everything.
> > > >    Maybe it would make sense to include this statement somewhere to make your contributions stand out more obviously (and avoid unnecessary discussion with ignorant reviewers).
> > > >    It is also my fault for not reading appendix C.2.1 of the revision, because there it is made perfectly clear that $\mathbf{F}$ is not invertible for $N_i \neq N_o$ and that this approach accounts for that.
> > > > 2. > We conjecture that the solution remains unique, and are aiming to derive a proof
> > > >
> > > >    I would (carefully) conjecture that the solution can **not** be unique.
> > > >    After all, I do not see how $\mathbf{M}$ could be unique if both $\tilde{\mathbf{U}}_\bot$ and $\tilde{\mathbf{V}}_\bot$ are not unique.
> > > >    I would argue that it would be useful to highlight this in the camera ready version (or the proof if I happen to be wrong again).
> > > > 3. > if the starting weights $\mathbf{W}_1 = \mathbf{A} \mathbf{V}^\mathsf{T}$ for non-diagonal $\mathbf{A}$, then the SVD of $\mathbf{W}_1$ will potentially not have $\mathbf{V}^\mathsf{T}$ as the right singular vectors.
> > > >
> > > >    Of course, that is another oversight from my side, thank you again for pointing this out.
> > > >    Therefore, the orthogonalisation I had in mind while writing the comment does not make sense at all (I was mainly thinking about the $\mathbf{R}$ matrices from [17]) and alignment might not be that bad after all.
> > > >    However, to really highlight this difference (and make it more obvious), it might be useful to include a statement along these lines in the main text.
> > > > 4. > each pair of singular values evolves with the shape of a sigmoidal, giving rise to the step-like learning dynamics
> > > >
> > > >    You are right that these step-like learning dynamics can also be found (in one figure) in [17].
> > > >    However, these dynamics are not really highlighted in any way.
> > > >    I also do not recall observing anything like this when training non-linear networks.
> > > >    I also unsuccessfully skimmed [25] for a statement along the lines of
> > > >    > step-like learning dynamics [are] considered an integral part of rich feature learning.
> > > >    However, I am not entirely familiar with this line of research and I do not think it is that important.
> > > >
> > > > PS: the code, which was included in the original supplementary, is not available anymore in the current revision.

---

> > > > > ### Author Response · Authors · 2022-08-09
> > > > > **Thank you for your thorough review**
> > > > >
> > > > > We thank the reviewer for the careful read of our paper and the many relevant reviews, we very much appreciate the long discussion and effort! It has improved the paper.
> > > > >
> > > > > We will clarify the first point in the main text to highlight our contribution.
> > > > >
> > > > > We will also work toward proof of the uniqueness of our solution. Briefly, the freedom of choice in $U_\perp$ and $V_\perp$ may not reflect in $QQ^T$ or $MM^T$. Regardless of the outcome, we will clearly state in the camera-ready version the result we find.
> > > > >
> > > > > Finally, we will incorporate in the latest version the clarification around the alignment dynamics.
> > > > >
> > > > > We are sorry for this mistake on our part, that we did not upload the code again. We appear unable to upload our code at the moment and will make sure it will be available again as soon as possible.

---

> ### Author Response · Authors · 2022-08-02
> **Answer 2 to Reviewer rRZa**
>
> > Is Assumption 3.1 an actual assumption? To me, this implies that the results only hold if, which should not be necessary if I understand it correctly. I think it would be better to simply state that assumption 2.1 is not needed anymore.
>
> Thank you, we now state that assumption 2.1 is not needed for Thm 3.1.
>
> > I would argue that assumption 3.3 is not necessary because it is implied by assumption 3.2
>
> Thank you for the careful read, we have dropped assumption 3.3, and mention that 3.2 implies no bottlenecks as a consequence.
>
> > Also, sufficient rank is a term I am not familiar with. I believe rectangular matrices are still said to be of full rank if their rank is the minimum of its dimensions. In this sense, Assumptions 2.4 and 3.2 are equivalent.
>
> We now use `full rank' consistently.
>
> > The proof of theorem 3.1 introduces a limit, that (I believe) is unnecessary. This leads to additional terms that should always be zero, making the entire formulation harder to understand.
>
> We now provide a rigorous proof of equation 8/9 (11/12 in the revised manuscript) in Appendix C (please see our earlier comment), by directly substituting our claimed solution into the gradient flow equations and verifying it. Hence the additional terms are necessary, and a key contribution of the paper. This direct approach sidesteps the limit argument entirely. Because the direct approach does not provide intuition on the form of the solution, we leave for the reader the revised non-rigorous derivation (Appendix C.2). We now further explain the (non-rigorous) rationale for this approach. In essence, similarly to how the pseudoinverse can be derived from the inverse as a limit, we study the behaviour of the zero eigenvalues of $F$ by considering a limit. We emphasise that this technique was how we came to the solution, but is not a rigorous proof.
>
> > I do not understand what the RSA is. The reference to [49] mentions multiple steps involved in the analysis. However, in the text it is used as something that should be straightforward to compute. Figure 4A seems to indicate it should be related to the covariance, but I do not know if this intuition would be correct.
>
> We have now revised and renamed the definition and discussion of the RSA. We compute aspects of the network’s internal representation by computing the task relevant representational similarity matrix (RSM) of the hidden layer. Reference [49] presents what is commonly thought to be the equivalent quantity in the field of neuroscience computed from neural measurements and introduces methods to compare representational similarities, e.g. acquired from an artificial neural network and functional magnetic imaging. In machine learning the representational similarity matrix would be referred to as the kernel matrix $\phi(x)^T\phi(x')$) of the neural representations in the hidden layer. We have revised the manuscript accordingly.
>
> > I fail to understand what it means for a solution to be rich or lazy. After all, rich and lazy are used to describe dynamics.
>
> We are proposing a more careful definition of the rich and lazy regimes. Rich and lazy have been used to describe both dynamics and internal representations. The literature introduced rich learning to describe non-linear dynamics that lead to task-specific internal representations. On the other hand, the lazy regime describes exponential dynamics and non-task-specific internal representations (that is, an NTK that remains unchanged over learning). Our paper attempts to separate these dynamical and representational aspects of the terms. We show that both exponential and non-linear dynamics can lead to task-specific solutions with favourable generalisation properties. Therefore, we disentangle the dynamics from the nature of the internal representation at convergence. We call rich representations those that are task-specific, often with favourable generalisation properties. We show that initialisation conditions can control the dynamics independently from the representation.
>
> > Figure 5C introduces target matrices, which are not explained in the text. I would have assumed that these are the targets for $ATA$, but then the off-diagonal elements should converge to non-zero values.
>
> Thank you for catching this error. The target matrices are ${\bf \tilde{U}\tilde{S}\tilde{V}^T}$. Hence, the targets for ${\bf A}^T{\bf A}={\bf S}$. We have corrected this error in the revision.
>
> > on line 11: the slow dynamics are described to be step-like, in contrast to exponential. However, the dynamics in figure 4E are still exponential (albeit after a plateau) and I would describe the dynamics in figure 5A as sigmoidal.
>
> The step-like behaviour is not clearly shown due to the log scale on the x axis. We have now adapted the figure accordingly.
>
> (Answer continued in comment below ...)

---

> > ### Comment · Reviewer_rRZa · 2022-08-03
> > **further questions/comments (part 2)**
> >
> > 1. > mention that 3.2 implies no bottlenecks as a consequence.
> >
> >    I think it would be nicer to be included as a general statement, since it also applies to assumption 1.4.
> >
> > 2. > Hence the additional terms are necessary
> >
> >    I do not see how the proof implies that the additional terms are necessary.
> >    After all, the proof does not show that the solution is unique and it seems like the original solution (and my proposed solution) are also provably solutions.
> >
> > 3. > The step-like behaviour is not clearly shown due to the log scale on the x axis. We have now adapted the figure accordingly.
> >
> >    I see now what you mean.
> >    However, why is this statement in the abstract if it is only observable/displayed for this toy example?
> >
> > (Answer continued in reply to comment above... )

---

> > > ### Author Response · Authors · 2022-08-07
> > > **Answer to further questions / comments (part 2)**
> > >
> > > > I think it would be nicer to be included as a general statement, since it also applies to assumption 1.4.
> > >
> > > We will update the camera-ready version accordingly.
> > >
> > > > I do not see how the proof implies that the additional terms are necessary. After all, the proof does not show that the solution is unique and it seems like the original solution (and my proposed solution) are also provably solutions.
> > >
> > > We provide a longer answer to this claim in the section above ("Answer to further questions / comments (part 3)"). The solution provided by the reviewer does not work as it has an error in the diagonalisation of ${\bf F}$.
> > >
> > > > I see now what you mean. However, why is this statement in the abstract if it is only observable/displayed for this toy example?
> > >
> > > If the network is initialised from (very) small random weights or from weights that share the left and right singular vectors with the task, the learning dynamics are only dependent on the difference of their singular values [17]. The temporal dynamics for each pair of singular values evolves with the shape of a sigmoidal, giving rise to the step-like learning dynamics [17]. This is independent of the task and happens generically--it does not only occur in this example. As starting from small initial weights was thought to be required for rich feature learning, step-like learning dynamics were considered an integral part of rich feature learning [17, 25]. Here we show that the two are in fact dissociated.

---

> ### Author Response · Authors · 2022-08-02
> **Answer 3 to Reviewer rRZa**
>
> > Would it be possible to generalise this analysis to deeper networks?
>
> The matrix Riccati equation approach cannot be straightforwardly generalised to deeper networks. However, it illuminates the dependence of dynamics on more complex initialisation schemes, providing novel insight into settings such as continual learning. In the longer term, we hope our results will be prerequisites for deeper extensions, though this is nontrivial.
>
> > Can you provide sufficient conditions for the invertibility of B?
>
> We initially thought we had found interpretable conditions, but we no longer believe we can and have removed those claims from the main text. Our results now simply assume the invertibility of ${\bf B}$, which is minimally restrictive — all of the results, insights, and observations presented in this paper hold under this assumption. We rely on the empirical observation that in the case of unequal input-output dimension B is always invertible, apart from the reversal learning case. Moreover, we exhibit an example case for which B is not invertible. We conjecture that only reversal learning scenarios can cause B to be singular, however, we are unable to systematically prove it.
>
> > Could you verify that equations 7 and 8 could be simplified by using ....
>
> We note that this simpler form only obtains for equal input-output size. We give a simplified solution for equal input-output size such that
> \begin{align}
> 	  & & \frac{1}{2}\begin{bmatrix}
> 	  {{\bf \tilde V}} \left(e^{{{\bf \tilde S}} \frac{t}{\tau}}{\bf B}^T -   e^{-{{\bf \tilde S}} \frac{t}{\tau}} {\bf C}^T\right)\\
> 	   {{\bf \tilde U}} \left(e^{{{\bf \tilde S}} \frac{t}{\tau}}{\bf B}^T +   e^{-{{\bf \tilde S}} \frac{t}{\tau}}{\bf C}^T\right)
> 	  \end{bmatrix} \\
> 	  &&\times\left[{\bf A}(0)^{-1} + \frac{1}{4}
> 	\left({\bf B}\left(\ e^{{{\bf \tilde S}} \frac{2t}{\tau}} - {\bf I}  \right){\bf {{\bf  S}}}^{-1}{\bf B}^T - {\bf C}\left(\ e^{-{{\bf \tilde S}} \frac{2t}{\tau}} - {\bf I}  \right){\bf {{\bf  S}}}^{-1}{\bf C}^T \right) \right]^{-1} \\
> 	&&\times\frac{1}{2}\begin{bmatrix}
> 	  {{\bf \tilde V}} \left(e^{{{\bf \tilde S}} \frac{t}{\tau}}{\bf B}^T -   e^{-{{\bf \tilde S}} \frac{t}{\tau}} {\bf C}^T\right)\\
> 	   {{\bf \tilde U}} \left(e^{{{\bf \tilde S}} \frac{t}{\tau}}{\bf B}^T +   e^{-{{\bf \tilde S}} \frac{t}{\tau}}{\bf C}^T\right)
> 	  \end{bmatrix}^T \\
> \end{align}
>
> This solution seems to go in the direction that was intended by your solution. For unequal input-output size, the additional terms in our solution are necessary.

---

> > ### Comment · Reviewer_rRZa · 2022-08-03
> > **further questions/comments (part 3)**
> >
> > 1. > cannot be straightforwardly generalised to deeper networks
> >
> >    Is there a non-straightforward way? E.g. recursively applying the analysis?
> >
> > 2. > We note that this simpler form only obtains for equal input-output size.
> >
> >    My derivation does not assume equal input-output size in any way.
> >    I simply noticed that $\mathbf{F}$ can be diagonalised without the need for limits, i.e.
> >     $$\mathbf{F} = \frac{1}{2}\begin{bmatrix}\tilde{\mathbf{V}} & \tilde{\mathbf{V}} \\\\ \tilde{\mathbf{U}} & -\tilde{\mathbf{U}}\end{bmatrix}
> >      \begin{bmatrix}\tilde{\mathbf{S}} & \mathbf{0} \\\\ \mathbf{0} & -\tilde{\mathbf{S}}\end{bmatrix}
> >      \begin{bmatrix}\tilde{\mathbf{V}}^\mathsf{T} & \tilde{\mathbf{U}}^\mathsf{T} \\\\ \tilde{\mathbf{V}}^\mathsf{T} & -\tilde{\mathbf{U}}^\mathsf{T}\end{bmatrix} = \frac{1}{2} \begin{bmatrix}\tilde{\mathbf{V}} \tilde{\mathbf{S}} \tilde{\mathbf{V}}^\mathsf{T} - \tilde{\mathbf{V}} \tilde{\mathbf{S}} \tilde{\mathbf{V}}^\mathsf{T} & \tilde{\mathbf{V}} \tilde{\mathbf{S}} \tilde{\mathbf{U}}^\mathsf{T} + \tilde{\mathbf{V}} \tilde{\mathbf{S}} \tilde{\mathbf{U}}^\mathsf{T} \\\\ \tilde{\mathbf{U}} \tilde{\mathbf{S}} \tilde{\mathbf{V}}^\mathsf{T} + \tilde{\mathbf{U}} \tilde{\mathbf{S}} \tilde{\mathbf{V}}^\mathsf{T} & \tilde{\mathbf{U}} \tilde{\mathbf{S}} \tilde{\mathbf{U}}^\mathsf{T} - \tilde{\mathbf{U}} \tilde{\mathbf{S}} \tilde{\mathbf{U}}^\mathsf{T}\end{bmatrix}$$
> >    Since this formulation also fits in your proof, I would assume that the formulation can be strongly simplified.
> >    Also, I believe that this reformulation (or the starting point for the proof) by itself might provide interesting insights and could be included in the main text.
> >    Additionally, this does not require invertibility of $\mathbf{B}$, making it clear(er) that invertibility is only needed for numerical stability.

---

> > > ### Author Response · Authors · 2022-08-07
> > > **Answer to further questions / comments (part 3)**
> > >
> > > > Is there a non-straightforward way? E.g. recursively applying the analysis?
> > >
> > > For each layer added to the neural network, the system is increasing by an additional non-linear differential equation. We are not aware of an approach to form a quantity equivalent to the ${\bf Q}{\bf Q}^T$ matrix with known solution for higher order systems and a study of such systems is beyond the scope of this work. We still hope there will be a non-straightforward way, and that our work will form a useful prerequisite.
> > >
> > > > My derivation does not assume equal input-output size in any way. I simply noticed that ${\bf F}$ can be diagonalised without the need for limits ...
> > >
> > > We would like to thank the reviewer for spending so much time verifying our equations. However, we first note that for unequal input-output dimensions
> > > $${\bf F} = \frac{1}{2}
> > > \begin{bmatrix}
> > > {\bf \tilde{V}} & {\bf \tilde{V}}\\\\
> > > {\bf \tilde{U}} & -{\bf \tilde{U}}
> > > \end{bmatrix}
> > > \begin{bmatrix}
> > > {\bf \tilde{S}} & 0\\\\
> > > 0 & -{\bf \tilde{S}}
> > > \end{bmatrix}
> > > \begin{bmatrix}
> > > {\bf \tilde{V}} & {\bf \tilde{V}}\\\\
> > > {\bf \tilde{U}} & -{\bf \tilde{U}}
> > > \end{bmatrix}^T
> > > = {\bf O}{\bf \Lambda}{\bf O}$$
> > >
> > > is not a diagonalisation of ${\bf F}$ as $O$ is non-square -- and while $O^TO = I$ -- crucially $OO^T \neq I$. More generally, if $N_i < N_o$ then ${\bf \tilde{U}} \in \mathbb{R}^{N_o \times N_i}$ and ${\bf \tilde{V}} \in \mathbb{R}^{N_i \times N_i}$ and therefore ${\bf O} \in \mathbb{R}^{(N_i+N_o) \times (N_i + N_i)}$.
> > > As a consequence, the solution provided by the reviewer in the earlier comment is erroneous.
> > >
> > > In our work, we use the diagonalisation
> > > $${\bf F} = \frac{1}{2}
> > >     \begin{bmatrix}
> > >         {\bf \tilde{V}} & {\bf \tilde{V}} & {\bf \tilde{V}_\perp}\\\\
> > >         {\bf \tilde{U}} & -{\bf \tilde{U}} & {\bf \tilde{U}_\perp}
> > >     \end{bmatrix}
> > >     \begin{bmatrix}
> > >         {\bf \tilde{S}} & 0 & 0\\\\
> > >         0 & -{\bf \tilde{S}} & 0\\\\
> > >         0 & 0 & 0
> > >     \end{bmatrix}
> > >     \begin{bmatrix}
> > >         {\bf \tilde{V}} & {\bf \tilde{V}} & {\bf \tilde{V}_\perp}\\\\
> > >         {\bf \tilde{U}} & -{\bf \tilde{U}} & {\bf \tilde{U}_\perp}
> > >     \end{bmatrix}^T,
> > > $$
> > > where ${{\bf O}}$ is a square matrix of dimension  $(N_i+N_o) \times (N_i+N_o)$. This diagonalisation requires the inclusion of zero eigenvalues from which the additional terms originate. We emphasize that, under our assumptions, we are guaranteed that $F$ is nonsingular when $N_i=N_o$, and guaranteed that it is singular when $N_i\neq N_o$. We have aimed at making this as explicit as possible in the manuscript. If the reviewer has suggestions how the manuscript could be improved to emphasise this more, we would be happy to incorporate it into the manuscript. Finally, we note that it can also be easily verified numerically (with the code provided) that the additional terms are required.
> > >
> > > We agree with highlighting that inversion of ${\bf B}$ is only required for numerical stability, i.e., to have only vanishing time dependent terms. We have added a clarifying sentence to the manuscript and consider adding the equation to the camera ready versions (using the 10th content page).

---

### Official Review · Reviewer_Rrbu · 2022-07-10

**Rating:** 7
**Confidence:** 3
**Soundness:** 3 good
**Presentation:** 3 good
**Contribution:** 3 good

**Summary:**

The paper studies the training dynamics of deep (2-layer) linear networks. It derives the full gradient flow dynamics under mild conditions, in a form that is suitable for numerical evaluation. It then applies these results to study several deep learning phenomena including alignment, catastrophic forgetting, and knowledge revision.

**Questions:**

My main comment is that I do not understand parts of the proof of Theorem 3.1, and have listed several questions below. Looking at the code, these issues seem to be resolved (in particular, SVD computes both $\tilde{U}$ and $\tilde{U}_\perp$ etc.) which makes me think that they have more to do with how the theorem and proof are presented and do not indicate an actual flaw with the proof. I will be happy to revise my score if the authors can answer these questions and clarify the presentation in the paper.

1. Line 123: "We note that if $N_i < N_o$, the left and right singular vectors are of shape $N_o \times N_i$ and $N_i \times N_i$". If $A$ is a matrix with dimensions $N_o \times N_i$, its singular value decomposition is given by $A = U S V^T$ where $U$ has dimensions $N_o \times N_o$, and $V$ has dimensions $N_i \times N_i$. In other words, the left and right singular vector matrices are square, and so I don't understand the statement in the paper. Also, the matrix $S$ has the same dimensions as $A$ and is not square. Much of the proof revolves around completing partial bases to full ones, but it is not clear to me what this means, given that the singular vectors already span their respective spaces.

2. In the proof (Appendix B.1), equation (26) implies that $e^{F t/\tau}$ depends on $V_\perp$ and $U_\perp$ through $M$. However, according to (20), $F$ itself does not depend on these matrices, and therefore its exponent cannot depend on them either. This is clearly true in eq. (24), since if we take the limit we see that this dependence on $M$ disappears. However, in going to (25) it seems the $\epsilon$ limit was exchanged with the limit in the exponential series, leading to this result. (I believe the issue here is related to the previous one, as $F$ should already depend on complete bases obtained through SVD.)

**Limitations:**

This work has no obvious societal impact.

**Strengths And Weaknesses:**

[Score updated after authors addressed all of my questions]

The paper improves on previous works in that is presents a closed-form expression for the gradient flow of deep linear networks that works under fairly mild assumptions, and that can be readily evaluated numerically. In particular, equations (7-8) do not include any non-trivial matrix exponentials (i.e. exponentials that include non-diagonal matrices), and only involve decaying exponents (which are numerically stable).

The numerical experiments confirm the results presented in the paper. The applications of the theoretical results to alignment, catastrophic forgetting, and other effects show the promise that these results hold in shedding some light on interesting deep learning phenomena. The main limitation of this work is that it applies to deep linear networks, and it is not clear to what extent these results will extend to non-linear networks.

---

> ### Author Response · Authors · 2022-08-02
> **Answer to Reviewer Rrbu**
>
> We thank the reviewer for the insightful comments and careful reading. We have corrected all minor comments in the newly submitted version of the manuscript.
>
> > Line 123: "We note that if $N_i < N_o$, the left and right singular vectors are of shape $N_o \times N_i$ and $N_i \times N_i$." If $A$ is a matrix with dimensions $N_o \times  N_i $, its singular value decomposition is given by $A = USV^T$ where $U$ has dimensions $N_o \times N_o$, and V has dimensions $N_i \times N_i$. In other words, the left and right singular vector matrices are square, so I don't understand the statement in the paper. Also, the matrix has the same dimensions as and is not square. Much of the proof revolves around completing partial bases to full ones, but it is not clear to me what this means, given that the singular vectors already span their respective spaces.
>
> We have used the compact SVD in this paper (also known as the "reduced", "thin" or "economy-sized" SVD). That is, if ${\bf A}$ is a matrix with dimensions $N_o \times N_i$ where $N_i < N_o$ , its compact/reduced/thin/economy singular value decomposition is given by ${\bf USV^T}$  where ${\bf U}$  has dimensions $N_o \times N_i $, ${\bf S}$ has dimensions $N_i\times N_i$ (i.e. a square matrix with ordered, and only non-zero eigenvalues on its diagonal), and ${\bf V}$, has dimensions $N_i\times N_i$. It is formed by retaining only the first $N_i$ columns of ${\bf U}$. We now state this clearly in the revision. Initially, we thought the compact SVD would lead to clearer equations; however, following up on your comment, we are currently reviewing if using the full SVD would be more parsimonious and clear.
>
> > In the proof (Appendix B.1), equation (26) implies that $e^{\frac{{\bf F}t}{\tau}}$ depends on ${\bf U_{\perp}} $and ${\bf V_{\perp}} $ through ${\bf MM}^T$. However, according to (20), $F$ itself does not depend on these matrices, and therefore its exponent cannot depend on them either. This is clearly true in eq. (24), since if we take the limit we see that this dependence on $M$ disappears. However, in going to (25) it seems the $\epsilon$ limit was exchanged with the limit in the exponential series, leading to this result. (I believe the issue here is related to the previous one, as $F$ should already depend on complete bases obtained through SVD.)
>
> Thank you for the close reading, in light of this comment we have added a direct proof of equation 8/9 (11/12 in the revised manuscript) in Appendix C. That is, we substitute the claimed solution (Eqaution 8/9, 11/12 respectively) into the differential equation and directly verify it. This completely sidesteps the limit-based arguments, which we agree were non-rigorous. We believe establishing this solution for unequal input/output is a key technical result in the paper. However, this direct strategy does not provide insight into how we obtained the form of this solution.
>
> We therefore leave for the reader the non-rigorous derivation which we have extensively revised and explained in more detail (Appendix C.2) to provide intuition on how we got to the result. However, if reviewers judge this does not add to the paper we will remove it. In response to the specific question, the matrix exponential depends even on zero eigenvalues (because $e^0=1$) and so forming the full diagonalisation is necessary (even though the zero eigenvalues are not necessary to represent $F$). This point is independent from the limits involved. Regarding the limits, in a similar manner to how a pseudoinverse can be obtained as the limit of the inverse plus an infinitesimal diagonal matrix, we add infinitesimal values and take the limit. This approach is not rigorous, but it is how we arrived at the form of the solution. We have revised our notation and now discuss this in more detail in the revised manuscript in Appendix C.2.

---

> > ### Comment · Reviewer_Rrbu · 2022-08-09
> > **Reviewer response**
> >
> > I would like to thank the authors for addressing all of my questions. I reviewed the revised paper and appendix and the results and proof are all clear to me now. I will revise the score accordingly.

---

### Official Review · Reviewer_zYQs · 2022-07-17

**Rating:** 7
**Confidence:** 2
**Soundness:** 3 good
**Presentation:** 4 excellent
**Contribution:** 3 good

**Summary:**

The paper studies the impact of initialization settings on gradient based learning and describes a mathematical toolkit for two layer linear networks for understanding the impact of prior knowledge on learning dynamics. The paper derives the exact solutions to these dynamics in the two layer linear network regime as a function of network initialization, utilizing the Matrix Riccati solution.

**Questions:**

None.

**Limitations:**

As you point out in your summary, "While our results broaden the class of two-layer linear network problems that can be described analytically, they remain limited and rely on a set of assumptions", the paper by itself may come across as too restricted. However, I applaud stating this fact, as it makes it clear that the paper should be better considered as a first step in a direction that potentially gives tooling to analyze more generic network types.

**Strengths And Weaknesses:**

The paper is overall well written and a lot of effort has been placed in making the presentation of the topic very accessible. I specifically want to point out the high quality and informativeness of the included figures.

The paper continues a longer research direction that tries to understand learning dynamics in (linear) networks. By taking into account so called "rich learning regimes" the paper covers a larger class of settings (including pretraining settings hereby), which are also practically more relevant. A core contribution my understanding is picking up a older idea (Matrix Riccati formulation for learning dynamics) and re-establishing it in the deep learning setting.

---

> ### Author Response · Authors · 2022-08-02
> **Answer to Reviewer zYQS**
>
> We thank the reviewer for the kind comments on accessibility and clarity. We share the reviewer's hope that this work will open new doors to analysing more complicated networks in the future. We remain available for any further comments and questions.

---

### Meta-Review · Area_Chair_kJmp · 2022-08-26

**Recommendation:** Accept
**Confidence:** Certain

**Metareview:**

This paper derives the exact gradient flow dynamics in linear networks studying the impact of initialization under mild conditions so as to incorporate prior knowledge as initialization. Fukumizu's matrix Riccati formulation for learning dynamics is utilized for understanding deep linear networks. Under very mild conditions, the solution can be obtained numerically. The paper uses this solution to understand deep learning phenomena such as alignment, catastrophic forgetting and knowledge revision. In effect the authors provide the first step to provide mathematical tools for characterizing prior knowledge in neural networks.

Reviewers agree that while the formalism in the paper is limited to "linear" networks, the insights drawn from the analysis and research direction the paper opens up will be valuable for the deep learning community and bear interesting connections to different analyses. While initial evaluations were mixed, after the author's responses, all reviewers recommended acceptance for the paper.


**Award:**

No

---

### Decision · Program_Chairs · 2022-09-14

Accept